# CRAB: Cross-environment Agent Benchmark for Multimodal Language Model Agents

## Abstract

The development of autonomous agents increasingly relies on Multimodal Language Models (MLMs) to perform tasks described in natural language with GUI environments, such as websites, desktop computers, or mobile phones. Existing benchmarks for MLM agents in interactive environments are limited by their focus on a single environment, lack of detailed and generalized evaluation methods, and the complexities of constructing tasks and evaluators. To overcome these limitations, we introduce CRAB, the first agent benchmark framework designed to support cross-environment tasks, incorporating a graph-based fine-grained evaluation method and an efficient mechanism for task and evaluator construction. Our framework supports multiple devices and can be easily extended to any environment with a Python interface. Leveraging CRAB, we developed a cross-platform Crab Benchmark-v0 comprising 120 tasks in computer desktop and mobile phone environments. We evaluated 6 advanced MLMs using different single and multi-agent system configurations on this benchmark. The experimental results demonstrate that the single agent with GPT-4o achieves the best completion ratio of 38.01%.

## 1 Introduction

The development of autonomous agents for human-centric interactive systems—such as desktop OS (Zhang et al., a), websites (Zhou et al.; Koh et al.), smartphones (Zhang et al., b; Xing et al.), and games (Vinyals et al.; Wang et al., a)—has long been an important goal of AI research, aiming to convert natural language instructions into concrete operations. Traditionally, these challenges have been addressed using reinforcement learning (Mnih et al.). Recently, Large Language Models (LLMs) have demonstrated remarkable proficiency in natural language understanding and commonsense reasoning, making them vital tools for developing autonomous agents. This utility is further enhanced by Multimodal Language Models (MLMs), which improve the ability to interpret visual information from GUIs (Cheng et al.).

To effectively develop MLM-based autonomous agents for real-world applications, it is essential to create suitable benchmarks for standardized performance evaluation. However, existing benchmarks still have limitations in terms of interaction methods, platform diversity, evaluation metrics, static task dataset that prevent them from closely mirroring complex real-world applications. First, existing benchmarks that interact with the environments through pre-collected observation data from system environments (Sun et al.; Mialon et al.; Deng et al., 2023) fail to capture the dynamic nature of real-world scenarios without interactive exploration where data and conditions can change unpredictably. Second, existing benchmarks are typically evaluated on a single platform, either Web, Android, or Desktop OS (Shi et al., 2017; Xing et al.; Xie et al.). However, the practical applications usually involve tasks that span multiple platforms. For example, using a smartphone to take a photo and sending it to a desktop for editing with a graphics editor is a common real-world task across multiple platforms. Third, existing evaluation methods are generally either goal-based or trajectory-based (Shi et al., 2017; Xing et al.). Goal-based methods typically employ a coarse-grained binary reward, solely evaluating whether the final system state aligns with the task's objectives. In contrast, trajectory-based methods can offer more nuanced metrics by assessing the agent's actions against a gold trajectory yet ignore the possibility of multiple valid pathways to complete a task, making the evaluation results less fair. Lastly, task creation within these complex systems are not static and extensible with fixed templates (Sun et al.; Xie et al.), which limits the diversity and scope of tasks.

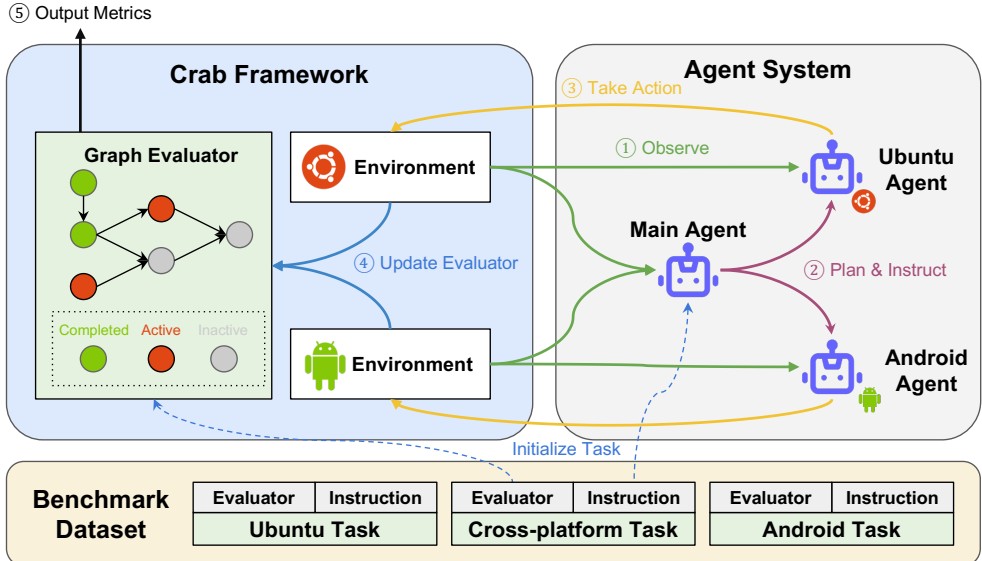

Figure 1: **Architecture of the Crab Framework demonstrating a benchmarking workflow for a multi-agent system.** A task is initialized by assigning instructions to the main agent and a graph evaluator inside the benchmark system. The workflow progresses through a cycle where the main agent observes, plans, and instructs the sub-agents, who then execute actions within their respective environments. The graph evaluator monitors the status of tasks within the environments, continuously updating and outputting the task completion metrics throughout the workflow.

We propose a benchmark that closely mirrors real-world situations and an evaluation method that more accurately reflects an agent's performance on complex tasks. To this end, we introduce CRAB, a novel **CR**oss-environment **A**gent **B**enchmark framework. CRAB provides a comprehensive framework for evaluating cross-environment tasks in interactive environments, where the agent needs to operate simultaneously across various devices and platforms, adapting to varied system conditions to complete tasks efficiently. To the best of our knowledge, CRAB is the first autonomous agent benchmark framework that incorporates the **cross-environment tasks**. Moreover, we propose a novel evaluation method called **graph evaluator**. Unlike traditional goal-based and trajectory-based evaluation, our graph evaluator checks the intermediate procedures of completing a task by decomposing the task into multiple sub-goals. Each sub-goal is assigned a judge function to verify its completeness, and each is considered a node in the graph evaluator. The graph structure describes the sequential and parallel relationships between the sub-goals. Therefore, it offers fine-grained metrics similar to trajectory-based evaluations while accommodating multiple valid pathways to a solution, making it more suitable for evaluating tasks that involve various correct approaches. To solve the increasing complexity in cross-environment task construction. We also propose a highly extensible graph-based task construction method called **sub-task composition**. Combining multiple sub-tasks in a graph with task targets allows for efficient construction of various cross-environment tasks with corresponding graph evaluators. The whole framework is implemented in Python and use the network to interact with environments, ensuring easy adaptation to any platform, device, or modality. Table 1 compares CRAB with existing agent benchmark frameworks.

Based on CRAB framework, we propose a benchmark Crab Benchmark-v0 with two cooperated environments that include an Android emulator and an Ubuntu desktop virtual machine. We have developed a total of 120 real-world tasks. These tasks address a wide array of common real-world applications and tools, including but not limited to calendars, email, maps, web browsers, and terminals, and facilitate common interactions between smartphones and desktops. Considerable time has been invested in verifying the accuracy and comprehensiveness of the instructions for subtasks, as well as the generalization and correctness of their evaluators. Most tasks are constructed using a careful composition of sub-tasks, while some tasks are crafted manually to accommodate specific collaborative scenarios. We test 6 popular MLMs, including GPT-4 Turbo, GPT-4o, Claude 3 Pro, Gemini 1.5 Pro, Pixtral-8B, and LLaVA-OneVision-72B across different structures of single-agent

Table 1: **Comparison of existing agent benchmark frameworks.** The columns details key features of each framework: *Interactive Environment* indicates the presence of either interactive environments or static datasets; *Multimodal Observation* specifies the availability of vision-based observations (e.g. screenshots); *Cross-platform* denotes support for multiple operating systems or platforms; *Evaluation* describes the evaluation metrics, categorized as *Goal-based* (checking environment state according solely on the final goal), *Trajectory-based* (comparing agent action trajectory with a gold actions sequence), *Multiple* (varied across tasks), *Intermediate-reward* (combines multiple signals with three strategies: Conjunctive Evaluation, Disjunctive Evaluation, and Order Constraint), *LLM-as-a-Judge* (Zheng et al., 2023), or *Graph-based* (a DAG with each node as an intermediate checkpoint); *Task Construction* shows the task construction method, including *Handmade* (handcrafted by human), *LLM-inspired* (using LLM to generate task drafts but still verified and annotated by human), *Template* (generated by filling in the blanks in task templates), or *Sub-task Composition* (composing multiple sub-tasks to construct tasks and evaluators).

| | Interactive Environment | Multimodal Observation | Cross-platform | Evaluation | Task Construction | # of apps or websites |
|---|---|---|---|---|---|---|
| MINIWOB++ (Shi et al., 2017) | Web | ✓ | ✗ | Goal-based | Handmade | 1 |
| WEBSHOP (Yao et al., 2022) | Web | ✓ | ✗ | Goal-based | Template | 1 |
| METAGUI (Sun et al.) | ✗ | ✗ | ✗ | Trajectory-based | Handmade | 6 |
| GAIA (Mialon et al.) | ✗ | ✗ | ✗ | Goal-based | Handmade | n/a |
| MIND2WEB (Deng et al., 2023) | ✗ | ✗ | ✗ | Goal-based | LLM-inspired | 137 |
| AGENTBENCH (Liu et al., 2024) | Multi-isolated | ✗ | ✗ | Multiple | Handmade | n/a |
| INTERCODE (Yang et al., b) | Code | ✗ | ✗ | Goal-based | Handmade | n/a |
| WEBARENA (Zhou et al., 2023) | Web | ✓ | ✗ | Goal-based | Template | 6 |
| OMNIACT (Kapoor et al.) | ✗ | ✗ | ✗ | Trajectory-based | Handmade | 60+ |
| VWEBARENA (Koh et al.) | Web | ✓ | ✗ | Goal-based | Template | 4 |
| ANDROIDARENA (Xing et al.) | Android | ✓ | ✗ | Trajectory-based | LLM-inspired | 9 |
| OSWORLD (Xie et al.) | Linux / Windows | ✓ | ✗ | Goal-based | Template | 9 |
| MOBILE-ENV (Zhang et al., 2024a) | Android | ✓ | ✗ | Intermediate-reward | Template | 13 |
| GUI-WORLD (Chen et al., 2024a) | ✗ | ✓ | ✗ | LLM-as-a-Judge | LLM-inspired | not present |
| ANDROIDWORLD (Rawles et al., 2024) | Android | ✓ | ✗ | Goal-based | Template | 20 |
| WAA (Bonatti et al., 2024) | Windows | ✓ | ✗ | Goal-based | Handmade | 6 |
| CRAB | Linux & Android | ✓ | ✓ | Graph-based | Sub-task Composition | 25 |

and multi-agent systems, totaling 12 different agent settings in our benchmarks. The experimental results show that the single agent structure with GPT-4o model achieves the best overall completion ratio of 38.01%, underscoring the necessity for ongoing development of more effective autonomous agents. Our proposed metrics successfully distinguish between different methods better than previous metrics. We further analyze the different termination reasons that reflect the problems inherent in the communication within the multi-agent system.

## 2 RELATED WORK

Leveraging LLMs as reasoning units has become an effective approach (Wang et al., 2024b; Huang et al., 2022; Xi et al.) for building autonomous agents, including embodied agents (Wang et al., a; Song et al., 2023; Chen et al., 2023), social simulations (Park et al., 2023; Lin et al., 2023), web navigation (Lù et al.), game playing (Lan et al., 2023; Tan et al., 2024), office assistants Li et al. (2024b), and code generation (Zhang et al., 2023). Specifically, some works apply LLMs to the planning of embodied agents in complex environments (Wang et al., a; Song et al., 2023; Chen et al., 2023). Others focus on simulating human behaviors and social communication by harnessing LLMs' remarkable human-like understanding and generation capabilities (Park et al., 2023; Lin et al., 2023). Additionally, multi-agent systems have been introduced to enhance the simulation of human behavior through agent cooperation (Li et al., 2023; Hong et al., 2023; Wu et al., 2023; Jin et al., 2024; Wang et al., 2024a). In another approach, several studies have expanded the capacities of agents by incorporating multimodal understanding, enabling agents to process diverse modalities of input data such as images and text (Hong et al.; Liu et al., a; Furuta et al., 2024; Chen et al., 2024b).

Various benchmarks are developed to validate the performance of autonomous agents based on the reproducible environments. Miniwob++ (Shi et al., 2017) analyzes the open-domain web tasks, builds corresponding web environment, and produces high-quality datasets considering extensive website and operation categories. Mind2Web (Deng et al., 2023) proposes a benchmark for the real-world websites which are genuine and unpredictable, with a high coverage of domains, websites, tasks, and user-interactions. WebArena (Zhou et al.) provides a realistic and reproducible web

environment to simulate sufficiently complex web tasks. Several works (Koh et al.; He et al., 2024) further broaden the web tasks, considering the visual tasks to build the benchmark for multi-modal autonomous agents. SWEBench (Jimenez et al.) builds a benchmark based on the Github, focusing on the coding capacity of understanding and solving issues. AgentBench (Liu et al., 2024) expands the scope of agent applications within the domain of computer interaction tasks and encompasses the examination of these tasks across multiple complex environments. OMNIACT (Kapoor et al.) incorporates the visual information of OS screen UI via segmentation and corresponding tagging, which creates corresponding tasks upon the basic elements. OSWorld(Xie et al.) pays attention to the simulations across diverse computer systems, taking XML and screenshots as both inputs and meticulously delineating a standardized format for both the environment and the evaluation process. WindowsAgentArena(Bonatti et al., 2024) focuses on the simulation of windows environment, proposes a challenging set of windows-oriented task, gives a trustful evaluation for the popular environment.

Current studies also focus on control tasks in mobile systems. MetaGUI (Sun et al.) divides the mobile system control tasks into dialogues and GUI operation traces, collecting GUI traces based on the collected dialogues. AITW (Rawles et al., 2023) produces a large dataset upon a large dataset of real-world scenarios, and builds challenging multi-steps tasks based on the annotated single-step tasks as a two-stage manner. MobileAgent (Wang et al., b) proposes tasks based on Ant Intelligent Assistant(AIA) system, which integrates Standing Operating Procedure(SOP) information for the creation of subtasks. AITZ (Zhang et al., 2024b) constructs datasets with Chain-of-Thought (COT) considerations, adding semantic annotations according to visual models at each step, and developing the operational procedure for selected tasks. Mobile Agent Bench (Wang et al., 2024c) collects app event signals via android accessibility service, builds the benchmark with well annotated operation trajectories, and divides the tasks into several levels. Android World (Rawles et al., 2024) establishes a fully functional environment for the Android system and provides a robust and reliable evaluation of the agent's capacity in Android-oriented tasks. Mobile-ENV (Zhang et al., 2024a) introduces an intermediate reward mechanism where the environment generates signals based on its state. These signals are combined into an intermediate reward using three types of aggregation operators. The motivation behind this approach aligns closely with the problem addressed by our proposed graph evaluator, but it relies on a tree structure with multiple relationships that can increase complexity for annotators, potentially limiting dataset scalability. GUI-World (Chen et al., 2024a) contributes a large video dataset for GUI automation and trains a new VideoLLM for UI tasks. However, the evaluation method used by GUI-World, LLM-as-a-Judger (Zheng et al., 2023), may lack the precision and consistency offered by rule-based evaluation systems.

## 3 DEFINITIONS

### 3.1 PROBLEM FORMULATION

Consider autonomous agents performing a task on a digital device (i.e. desktop computer). Such a device typically has input devices (i.e. mouse and keyboard) for human interaction and output devices (i.e. screen) to allow human observation of its state. In CRAB, we represent this type of device as an **environment**. Formally, this environment is defined as a reward-free Partially Observable Markov Decision Process (POMDP), denoted by the tuple $M := (\mathcal{S}, \mathcal{A}, \mathcal{T}, \mathcal{O})$, where $\mathcal{S}$ represents the state space, $\mathcal{A}$ the action space, $\mathcal{T} : \mathcal{S} \times \mathcal{A} \to \mathcal{S}$ the transition function, and $\mathcal{O}$ the observation space. Considering the collaborative nature of multiple devices in real-world scenarios, we can combine multiple environments into a set $\mathbf{M} = M_1, M_2, ..., M_n$, where $n$ is the number of environments and each environment $M_j = (\mathcal{S}_j, \mathcal{A}_j, \mathcal{T}_j, \mathcal{O}_j)$. We define a task that requires operations across multiple environments as a **cross-environment task**. This task is formalized as a tuple $(\mathbf{M}, I, R)$, in which $\mathbf{M}$ is the environment set, $I$ is the task objective in the form of natural language instructions, and $R$ is the reward function of the task. An **agent system**, designed to complete a task represented by an instruction $I$, can be modeled as a policy $\pi((m, a) \mid (I, H, o_1, ..., o_n))$, which defines the probability of taking action $a$ in environment $m$ when receiving observation $(o_1, ..., o_n)$ from environment $(M_1, ..., M_n)$ with a history of actions $H$. An **agent** within the agent system operates with a fixed back-end MLM, a predefined system prompt, and retains its chat history. An agent system is composed of either a single agent responsible for all planning, reasoning, and action-taking or multiple agents connected through a communication workflow to collaborate.

## 3.2 Graph of Task Decomposition

Decomposing a complex task into several simpler sub-tasks has been proved to be an effective prompting method for LLMs (Khot et al., 2023). Some studies represent sub-tasks in a graph structure. For instance, PLaG (Lin et al.) uses a graph-based structure to enhance plan reasoning within LLMs, while DyVal (Zhu et al., 2024) employs directed acyclic graphs (DAGs) to facilitate dynamic evaluation of LLMs. By introducing this concept into the realm of benchmarks, naturally, decomposing a complex task into sub-tasks that have both sequential and parallel connections forms a DAG. Therefore, we introduce the **Graph of Decomposed Tasks** (GDT), which provides a new task decomposition method representing decomposed sub-tasks within a DAG structure. In GDT, each node is a sub-task, formalized as a tuple $(m, i, r)$, where

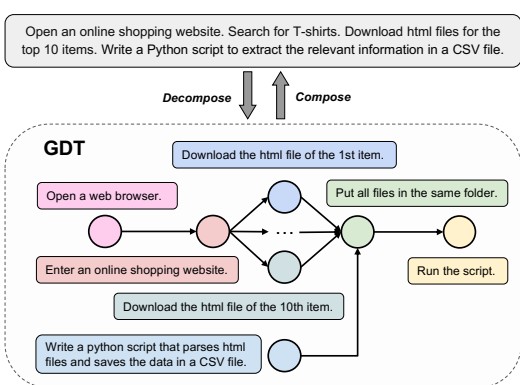

Figure 2: **An example of a Graph of Task Decomposition.**

$m$ specifies the environment in which the sub-task is performed, $i$ provides the natural language instruction, and $r$ represents the reward function. This function evaluates the state of $m$ and outputs a boolean value to determine if the sub-task is completed. The edges within GDT represent the sequential relationship between sub-tasks. An example GDT is shown in Fig. 2.

## 4 The Crab Framework

### 4.1 Cross-environment Agent Interaction

Compared to single-environment tasks, cross-environment tasks offer two main advantages for benchmarking agents. First, cross-environment tasks reflect real-world scenarios where humans use multiple devices simultaneously to accomplish tasks. Second, these tasks require sophisticated message processing and information transfer between environments. Such tasks demand that the agent plan actions, construct outputs for each environment, and remember what needs to be transferred, showcasing a high-level understanding of real-world and ability to solving complex tasks. CRAB uses a unified interface for agents to operate in all environments. We define an action by its name, the environment it belongs to, a concrete description of its functionality, and the parameters with descriptions. Through this approach, CRAB can adapt to any platform or modality, from devices to applications like browsers, by defining a few interactive functions. Implementation details are in the Appendix A.3.

### 4.2 Graph Evaluator

To assess the capabilities of MLM agents, most benchmarks (Shi et al., 2017; Deng et al., 2023; Koh et al.; Zhou et al.) evaluate agents based on solely the final states of the environment after agent operations. Typically, they only judge whether the final goal is success or fail. However, this approach does not capture incremental progress made by the agents. For instance, consider two agents tasked with installing a new application on a computer: agent $a$ successfully downloads the installer but fails during the installation process, whereas agent $b$ does not even try to find the installer. Despite Agent $a$ making more progress, both are deemed failures under the goal-based evaluation system, resulting in an unfair assessment of their performance. An alternative method, *Trajectory-based Matching* (Xing et al.; Kapoor et al.), abandons state-based evaluation and instead compares the agent's actions against a predefined gold action sequence for each task, giving nuanced metrics. Nevertheless, this method faces challenges in real-world systems where tasks may have multiple valid execution paths. For example, copying a file can be accomplished using either a file manager or the command line. Inspired by the "decomposing" idea from GDT (Sec. 3.2), we propose a novel integrated approach, the *Graph Evaluator*, which provides fine-grained metrics and supports multiple valid paths.

To build a graph evaluator for a given task, we begin by decomposing the task into a GDT, where each sub-task is associated with an intermediate environment state critical to completing the overall task. Nodes in the graph evaluator activate when they either have no incoming edges or after all their preceding tasks are completed, ensuring a sequential order of tasks. After an agent takes an action, the system checks these active nodes to verify if the target state of each node is reached. A node completion triggers successor nodes to activate and verify the state. This cycle repeats until no new nodes activate, showing that the system's task sequence aligns with the current state of the environment. Unlike trajectory-based methods, the Graph Evaluator focuses on **key states** rather than specific actions, allowing agents flexibility in execution. For instance, in a file-editing task, the evaluator checks if the file is edited, regardless of whether a CLI or GUI editor is used. This ensures mandatory steps are completed while accommodating diverse execution paths.

### 4.3 METRICS

Given a Graph Evaluator synchronized with the environment state, it becomes possible to track agent progress through the current status of sub-task completions. Beyond the traditional **Success Rate (SR)**, which marks a task as *success* only when all sub-tasks are completed, we introduce three metrics aiming at assessing both performance and efficiency of agents, leveraging the detailed sub-task status provided by the graph evaluator. Specifically, the **Completion Ratio (CR)** measures the proportion of completed sub-task nodes relative to the total nodes in the graph, calculated as $C / N$, where $C$ is the number of completed nodes and $N$ is the total number of nodes. This metric offers a straightforward measure of an agent's progress on a given task. The **Execution Efficiency (EE)**, calculated as CR $/ A$, where $A$ denotes the count of executed actions. It evaluates how efficiently actions are executed relative to the completion of nodes, reflecting the agent's task execution efficiency. Lastly, the **Cost Efficiency (CE)**, calculated as CR $/ T$, where $T$ is the total number of model tokens used, evaluates the efficiency of resource consuming by the agent.

### 4.4 TASK AND EVALUATOR CONSTRUCTION

Despite the graph evaluator offering detailed evaluations, its creation is complex, requiring task decomposition into sub-tasks with well-defined graph structures and expert involvement. To ease task and evaluator creation, we propose building GDTs by sub-tasks, addressing two main challenges: (1) the need for manual creation of sub-tasks and (2) the complexity of modeling sequential and parallel relationships between them. A template-based approach is commonly used to address the first issue by generating a large number of tasks efficiently. To tackle the second challenge, we employ the message transferring concept (Sec. 4.1). Specifically, if a sub-task $\alpha$ produces an output message that serves as an input for another sub-task $\beta$, then $\alpha$ can be considered a legitimate prerequisite of $\beta$, allowing us to connect $\alpha$ and $\beta$ with an directed edge in the GDT. To further refine our approach, we introduce a *sub-task template* structure. Each sub-task is described using a natural language instruction template that includes several replaceable input attributes. The types of each input attribute and the task output should be defined carefully. To generate a GDT, input attributes can be filled with either a hand-crafted value corresponding to their type or linked to a task with the same output type as the input type. From the evaluator's perspective, each sub-task template is linked to an evaluator generator that uses the input attribute value to generate evaluator subgraphs. Once a GDT is constructed, the graph evaluator is created by interlinking each subgraph. We follow the principle that each subtask should **do one thing within a single environment**, with clearly defined inputs and outputs that enable seamless integration with other tasks. For example, downloading a file from a URL to a file path is a well-defined subtask: it accepts a URL as input and outputs the file's contents.

Task descriptions are initially generated by GPT-4 from sub-task prompts and refined by human reviewers. This approach, unlike naive templates, allows for a more detailed and scalable task composition. Our method automates graph evaluator generation, relieving users of coding requirements and making the system accessible to a broader audience.

## 5 THE CRAB BENCHMARK

**Environments.** We build an agent benchmark Crab Benchmark-v0 featuring with cross-environment, graph evaluator, and task generation through CRAB framework. The environments consists of an Android smartphone emulator and a Ubuntu Linux desktop virtual machine. We establish both

environments in a reproducible and standalone manner and utilize snapshots to ensure a consistent initial state for all environments. The observation space consists solely of the current system screen for both environments, captured in image format at each step of the agent's interaction. We employ the Set-of-Marks visual prompt method (Yang et al., a) to label each interactive element on the screen. Interactive elements are identified using the GroundingDINO (Liu et al., b) with `icon.logo.` text prompt to locate all interactive icons. Additionally, Optical Character Recognition (OCR) is utilized through EasyOCR[1] to detect and label interactive text elements. Each detected item is assigned a unique integer ID, facilitating reference within the action space. The action spaces for Ubuntu and Android are distinct and designed to be close to the common interactions in the real devices. For Ubuntu, we define the following actions: mouse-based actions, keyboard-based actions and a shortcut action to search for applications. For Android, the action set includes tapping actions, a text action, a physical button action, and an action to open the app drawer. Additionally, we introduce three environment-irrelevant actions: completing the task, submitting an answer and waiting. Detailed descriptions for the environment implementation are shown in Appendix A.2.

**Tasks.** We meticulously construct 17 sub-task templates for the Android environment and 19 sub-task templates for the Ubuntu environment. The Ubuntu templates encompass a variety of tasks such as Command Line Interface (CLI) operations, file system management, search engine usage, desktop configurations, and map navigation. Conversely, the Android sub-task templates are primarily focused on the storage and transmission of messages via various applications. Each sub-task template is linked to a graph evaluator consisting of one to four nodes. Each sub-task are its graph evaluator is verified by at least two related field experts. We make sure that all tasks are reachable by human. We generate 104 tasks by sub-task composition and make 16 tasks by hand to include more complex scenarios that cannot easily described by the sub-tasks. The dataset has 29 Android tasks, 73 Ubuntu tasks and 18 cross-platform tasks, totaling 120 tasks. Our tasks are intentionally designed to be more complex than those in other benchmarks, which naturally requires more time for design and experimentation. A single sub-task in our benchmark might involves multiple operations across several applications, unlike prior works where most tasks often focus on solving problems within a single application. With multiple applications nature combined with the scalability of our task composition and graph evaluator, our tasks are sufficiently challenging to test an agent's performance across different applications and scenarios, thereby effectively assessing its generalization ability. The format and the applications covered by the dataset are shown in Appendix A.4 and A.5, respectively.

**Evaluators.** To assess the intermediate states of sub-tasks as described in Sec. 4.2, we have implemented a comprehensive suite of execution-based evaluators. These evaluators retrieve and assess specific current states, such as the edited content of a file or a modified setting, thereby determining the successful completion of a sub-task. For each evaluator, input attributes are carefully selected to interpret software information or system settings relevant to the scenario defined for the sub-task. For instance, evaluators use file paths before and after edits as input parameters to verify the completion of file editing sub-tasks. Specifically, for sub-tasks on the Android platform, we incorporate XML-based evaluators (Xing et al.). We dump UI layout as XML path and verify whether the UI content matches the expected state. For the Ubuntu platform, we employ image matching techniques (Potje et al., 2024; Jiang et al., 2024; Edstedt et al., 2024) and OCR to handle scenarios where acquiring necessary state information through conventional APIs is challenging. Image matching offers fine-grained visual correspondences by comparing keypoint features between images, allowing us to assess spatial relationships among visual elements. Using OCR and image matching, we can accurately evaluate tasks such as verifying whether an agent has successfully created a slide with specified images, text content, and layouts—tasks for which trivial evaluation methods are lacking. We utilize EasyOCR[1] and XFeat[2] as our primary tools for OCR and image matching. For tasks with real-time characteristics that may change over time, we implement crawler scripts to capture dynamic values at the moment of evaluation. These values are then compared with the results achieved by the agent upon task completion. We have a total of 59 evaluator functions with different types. Each task has 4.2 evaluators in average of the whole dataset.

---

[1] https://github.com/JaidedAI/EasyOCR
[2] https://github.com/verlab/accelerated_features

## 6 EXPERIMENTS

### 6.1 BASELINE AGENT SYSTEM

At the core of MLM Agents are backend Multimodal Language Models that provide natural language and image understanding, basic device knowledge, task planning, and logical reasoning abilities. To run in Crab Benchmark-v0, the backend model needs to support: (1) Accept multimodal mixed input, as the system provides both screenshots and text instructions as prompts; (2) Handle multi-turn conversations, as most tasks require the agent to take multiple actions, necessitating the storage of history messages in its context; (3) Generate structured output through function calling, ensuring the proper use of provided actions with type-correct parameters. However, most open source models do not provide explicit function calling feature, we let these models generate structured JSON output to simulate the function calling behavior.

We selected 4 commercial and 2 open source MLMs that meet these criteria for our experiments: GPT-4o (gpt-4o-2024-05-13) (OpenAI, 2024), GPT-4 Turbo (gpt-4-turbo-2024-04-09) (Achiam et al.), Gemini 1.5 Pro (May 2024 version) (Reid et al.), Claude 3 Opus (claude-3-opus-20240229) (Anthropic, Year), Pixtral-12B (Pixtral-12B-2409)[3], and LLaVA-OneVision-72B (llava-onevision-qwen2-72b-ov-chat) (Li et al., 2024a). These models serve as the backend models for our agents. Specifically, We use function calling feature in the four commercial models and JSON output in the two open source models that do not support function calling. Since the JSON output setting uses different prompts from the other, we employ a GPT-4o agent without function calling as the control group to the open source models.

Beyond the MLM backend, the structure of agent systems also influences overall performance. To examine how different multi-agent structures impact performance, we design three agent system structures: **single agent**, **multi-agent by functionality**, and **multi-agent by environment**. In the **single agent** structure, one agent manages all responsibilities, including observation analysis, planning, reasoning, and format the output action. The **multi-agent by functionality** structure splits tasks between a main agent, responsible for analysis and planning, and a tool agent that translates instructions into actions without accessing environmental observations. This division allows the main agent to concentrate on high-level tasks without managing functional call formats. Meanwhile, in the **multi-agent by environment** setup, responsibilities are further distributed. A main agent processes all environmental observations for high-level planning, while each environment-specific sub-agent executes actions based on the main agent's instructions, incorporating observations from their respective environments.

For all models, we utilized the default API parameters and retained two turns of historical messages to ensure messages do not exceed the context window. The interaction turns are limited to 15 and the task will be terminated when reaching max turns. The agent can also terminate the task ahead if it thinks the task is completed. The screenshots are passed through PNG format with the highest quality that the APIs provide. Detailed agent and prompt designs are shown in Appendix B. In the experiment, we deployed four cloud machines cloned from the same disk image to ensure a consistent environment for all agents. Running a single agent setting in the benchmark requires at least 30 hours to complete on one machine. Evaluation duration depends on the agent system, API response time, and task steps. Single-agent systems average 10 to 20 seconds per step, while multi-agent systems take 20 to 40 seconds.

### 6.2 RESULT

The primary outcomes are detailed in Table 2. Aside from the *Success Rate*, *Completion Rate*, *Execution Efficiency*, and *Cost Efficiency* mentioned above, we also present the reasons for agent termination to further investigate the factors preventing the agent system from completing the task.

**Comparison of backend models.** The GPT-4o and GPT-4 Turbo models, developed by OpenAI, achieved the highest average success rates and completion ratios (CR) among the tested models. Claude 3 outperforms Gemini 1.5 in terms of CR, but there remains a significant gap between the GPT-4 series and other models. Claude and Gemini have a higher Invalid Action Ratio, usually failing by clicking nonexistent elements on the screen or taking nonexistent actions. Regarding

---

[3]https://mistral.ai/news/pixtral-12b/

Table 2: **Evaluation results on Crab Benchmark-v0.** The *Model* column identifies the backend masked language models (MLMs) used. The *Structure* column describes the configuration of the agent system: *Single* means *single agent*; *By Func* is *multi-agent by functionality*; *By Env* indicates *multi-agent by environment*. We provide traditional metric of *Success Rate* (SR) alongside newly introduced metrics: *Completion Ratio* (CR), *Execution Efficiency* (EE), and *Cost Efficiency* (CE). Note that Gemini 1.5 Pro has an invalid CE because the Gemini API does not support retrieving token counts at the start time of experiments. The *Termination Reason* shows the ratio of reasons why the agent is terminated when the task is not success. *False Completion* (FC) indicates that the agent believes it has completed the task, but it actually has not; *Reach Step Limit* (RSL) means the agent has reached the step limit but has not completed the task; *Invalid Action* (IA) refers to the agent producing outputs that do not follow instructions, which may include invalid formats, nonexistent actions, or invalid action parameters.

| Agent system | | Metrics | | | | Termination Reason | | |
|---|---|---|---|---|---|---|---|---|
| Model | Structure | SR(%) ↑ | CR(%) ↑ | EE(%) ↑ | CE(%) ↑ | FC(%) | RSL(%) | IA(%) |
| GPT-4O | Single | 14.17 | **38.01** | **4.15** | $5.29 \times 10^{-4}$ | 8.33 | 55.83 | 21.67 |
| GPT-4O | By Func | **15.00** | 34.00 | 3.93 | $\mathbf{5.31} \times 10^{-4}$ | 10.83 | 54.17 | 20.00 |
| GPT-4O | By Env | 14.17 | 33.34 | 3.84 | $2.74 \times 10^{-4}$ | 8.33 | 48.33 | 29.17 |
| GPT-4 TURBO | Single | 9.17 | 33.35 | 3.80 | $4.52 \times 10^{-4}$ | 8.33 | 65.00 | 17.50 |
| GPT-4 TURBO | By Func | 13.33 | 33.48 | 4.07 | $4.38 \times 10^{-4}$ | 10.83 | 40.00 | 35.83 |
| GEMINI 1.5 PRO | Single | 5.00 | 15.48 | 1.72 | n/a | 2.50 | 55.83 | 36.67 |
| GEMINI 1.5 PRO | By Func | 5.00 | 12.76 | 1.42 | n/a | 8.33 | 33.33 | 53.33 |
| CLAUDE 3 OPUS | Single | 3.33 | 19.60 | 1.95 | $1.85 \times 10^{-4}$ | 10.00 | 57.50 | 29.17 |
| CLAUDE 3 OPUS | By Func | 3.33 | 16.48 | 1.72 | $1.77 \times 10^{-4}$ | 28.33 | 34.17 | 34.17 |
| GPT-4O W/O FC | Single | 9.17 | 23.05 | 2.34 | $3.93 \times 10^{-4}$ | 5.00 | 42.50 | 43.33 |
| PIXTRAL-12B | Single | 0.83 | 9.50 | 0.75 | $0.87 \times 10^{-4}$ | 0.83 | 75.83 | 22.50 |
| LLAVA-OV-72B | Single | 0.83 | 6.64 | 0.52 | $1.02 \times 10^{-4}$ | 12.50 | 71.67 | 15.00 |

efficiency, the GPT-4 series also demonstrates strong performance, with GPT-4o having a higher CE value compared to GPT-4 Turbo, highlighting its cost-effectiveness. GPT-4o's performance drops after disabling tool calling feauture, primarily due to its higher Invalid Action rate, showing the effectiveness of tool calling in generating structured output. In open source models, Pixtral-12B, with far fewer parameters, achieves a better CR compared to LLaVA-ov-72B, showcasing its efficiency. Although the open-source models generally understand screenshots and generate step-by-step plans correctly, they often fail to execute the correct actions according to the plan. Moreover, they do not effectively analyze task completion through observation. Once an incorrect action is performed, they tend to assume current step is success and proceed to the next step.

**Comparison of agent structures.** The performance of multi-agent structures on all backend MLMs is slightly lower than that of single-agent structures, which is somewhat unconventional. Based on the communication log, we find that multi-agent structures tend to experience information loss during inter-agent communication, leading to misunderstandings among downstream agents. This increases the likelihood of multi-agent structures taking invalid actions and incorrectly completing tasks. These experiments demonstrate that the design of the communication protocol and selecting the appropriate scenario are crucial for multi-agent systems. A detailed analysis is included in Appendix C.2. In terms of efficiency, multi-agent structures require more chat rounds, which can consume more tokens, resulting in a lower CE compared to single-agent settings.

**Comparison of platforms.** We have three types of tasks: Ubuntu, Android, and cross-environment. The metrics for each type of task can reveal the model or structure preferences. As shown in Fig. 3, the GPT-4o model demonstrates significantly better performance on Android and cross-platform tasks compared to GPT-4 Turbo, which may indicate OpenAI's increased focus on mobile devices. Additionally, models like Gemini, Claude, Pixtral, and LLaVa-OV perform better on Android devices compared to the Ubuntu, likely due to less training on Linux desktop data, which makes it difficult for them to recognize desktop icons. While it does not fully represent agent performance, it's notable that the two open-source models exhibit a low invalid action ratio but still fail to complete tasks. We include further platform specific results in Appendix C.1.

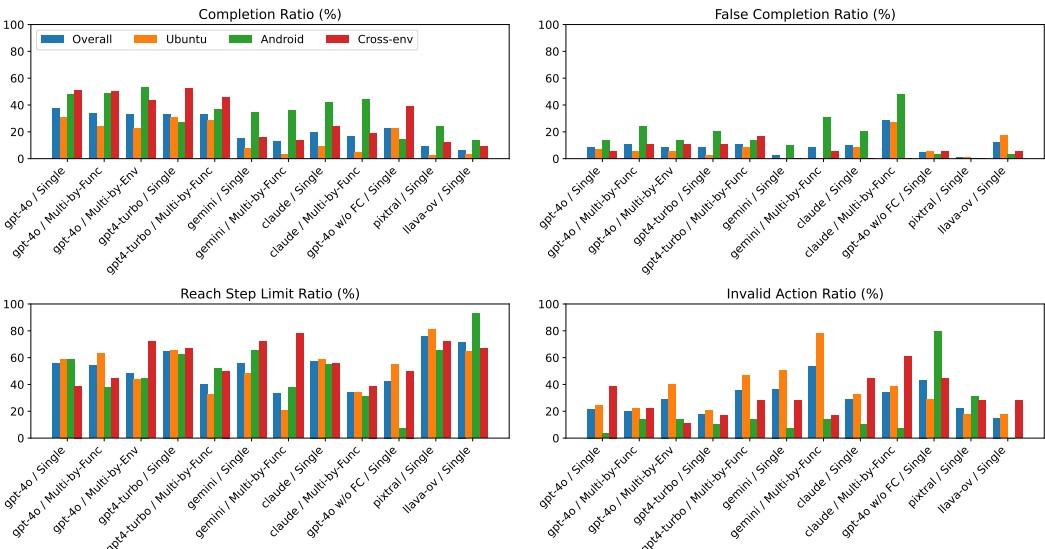

Figure 3: **Completion Ratio and Termination Reasons on different platforms.**

**Comparison of metrics.** The completion ratio metric reveals a notable performance difference between models. For instance, even though GPT-4o with single agent strcuture and with mutli-agent by environment structure have the same success rates, their completion ratios differ by up to 4.67%. This highlights the value of the completion ratio in assessing the effectiveness of different methods. For a more detailed analysis of each model and structure's performance, we provide several case studies in the Appendix. C.3.

**Key issues in solving cross-environment task.** The benchmark pipeline's complexity makes it difficult to identify universal issues across tasks and models. However, the challenges in cross-platform tasks are similar to those in single-platform settings. Key issues include **action space discrepancies**, where diverse action spaces in cross-platform environments confuse single-agent architectures but can be mitigated by multi-agent setups tailored to each platform; **limited context length**, which prevents the ability to process entire history observations and becomes more severe for cross-platform scenarios with increasing screenshots; **coordinate grounding issues**, where advanced tools like GroundingDINO and OCR occasionally fail to detect all screen elements in too complicated GUI observation; and **icon recognition failures**, where the backend model correctly plans the next step but cannot accurately identify and interact with corresponding icons, even though the visual prompt detect them correctly.

## 7 CONCLUSION

We propose the CRAB framework, which introduces the cross-environment automatic task-performing problem, featuring advanced graph-based task generation and evaluation methods that reduce manual effort in task design and provide more dynamic and accurate agent assessments. Based on this framework, we present Crab Benchmark-v0, a set of high-quality cross-environment tasks in smartphone and desktop environments, equipped with advanced visual prompting techniques. We tested various backend models and agent system structures on this dataset. The results reveal preferences for different agent settings, demonstrating Crab Benchmark-v0's strong ability to distinguish MLMs and autonomous agent systems. Despite our contribution to advancing cross-environment agent research, there are still some limitations. The sub-tasks are built upon the original apps in the Ubuntu and Android systems on Pixel devices, which limits the coverage of a wider range of applications. The current visual prompting methods do not fully recognize all interactive elements, hindering agent performance. Future work can focus on expanding the dataset and environments, testing more models, prompts, and multi-agent structures, as well as improving the use of visual prompting methods within the benchmark.

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

# A  BENCHMARK DETAIL

Section A.2 introduces the implementation details and action space settings of the benchmark environments. Section A.3 describes the design logic and implementation of the CRAB framework. Section A.4 describes the our experiment settings in detail. Section A.5 describes the specific format defined in our framework that ease data extension and how to use them. We provides a detailed document to setup experiment environments and reproduce our results.[4] Fig. 4 shows the structure of modules inside Crab Benchmark-v0.

Figure 4: **Module Structure of Crab Benchmark-v0.** The benchmark is divided into two primary sections: the left section, highlighted with warm hues, features two environments, while the right section, accentuated with cool hues, outlines various tasks. Each environment is defined by attributes including name, description, observation space, prompt method, and action space. Blocks marked in red denote actions. As for the tasks, they are composed of multiple sub-tasks and formulated by combine multiple evaluator sub-graphs derived from the sub-task evaluator generators. Arrows illustrate the compositional relationships between tasks and sub-tasks.

## A.1  DATASET STATISTICS

The applications in our task dataset along with the counts of tasks that utilize them is listed in Table 3 and 4. The task dataset covers a wide range of applications across two platforms, primarily focusing on daily life, programming, and office work scenarios. It is also worth noting that in our task settings, a single task often involves two or more applications. On average, each task contains 1.84 applications, according to our statistics.

Table 3: **Applications and their task counts in the Ubuntu environment.**

| App Name | Description | # Tasks |
|---|---|---|
| Terminal | GNOME terminal emulator with command line tools (e.g., cat, wget). | 40 |
| Firefox | Web browser with various web Apps (e.g., Google Docs and Search). | 35 |
| File Manager | GNOME official file manager. | 25 |
| GIMP | GNU Image Manipulation Program, open-source raster graphics editor. | 13 |
| System Setting | GNOME system setting GUI application. | 11 |
| VSCode | Code editor. | 8 |
| LibreOffice Writer | Word processor. | 8 |
| LibreOffice Impress | Presentation program. | 7 |
| LibreOffice Calc | Spreadsheet program. | 6 |
| Vim | CLI text editor. | 6 |
| Slack | Team communication platform. | 1 |

Table 4: **Applications and their task counts in the Android environment.**

| App Name | Description | # Tasks |
|---|---|---|
| Google Map | Map application. | 13 |
| Google Calendar | Calendar application. | 9 |
| Gmail | Google mail service application. | 7 |
| Google Keep | Google note application. | 6 |
| Google Tasks | Google TO-DO list. | 5 |
| Messages | Android built-in message sending application. | 5 |
| Contacts | Android built-in contacts application. | 5 |
| Google Drive | Google Cloud Drive application. | 4 |
| Clock | Android built-in clock application. | 2 |
| Files | Android built-in file manager. | 1 |
| Settings | Android system setting. | 1 |
| Camera | Android built-in camera. | 1 |
| Google Docs | Google online word processor. | 1 |
| Phone | Android built-in phone calling application. | 1 |

The distribution of node counts of graph evaluators per task is provided in Table 5. Our task dataset includes graphs ranging from 1 to 11 nodes. It is important to note that the number of nodes depends on the complexity of the task, with more complex tasks involving larger graphs.

A.2    ENVIRONMENT IMPLEMENTATION DETAIL

The Ubuntu environment is launched on a QEMU/KVM (Bellard, 2005; Kivity et al., 2007) Virtual Machine, and the Android environment employs the Google Android Emulator[5]. Interaction with the Ubuntu environment is facilitated using PyAutoGUI[6] and MSS[7], which provide high-level commands for mouse and keyboard control and screen capture, respectively. For the Android environment, we use the Android Debug Bridge (ADB)[8]. The detailed action space is described in Table 6.

A.3    FRAMEWORK DESIGN

CRAB offers a modular and extensible framework for evaluating agent performance in diverse tasks. At the heart of the framework lies the *action*, a unit operation representing the fundamental operation within the benchmark. The *action* is essentially an executable Python function that can be defined with explicit typed parameters and a clear description. *actions* serve not only as building blocks but

---

[4] https://github.com/camel-ai/crab/blob/main/crab-benchmark-v0/README.md

[5] https://developer.android.com/studio/run/emulator

[6] https://github.com/asweigart/pyautogui

[7] https://github.com/BoboTiG/python-mss

[8] https://developer.android.com/tools/adb

Table 5: **Node count histogram.**

| # Nodes | 1 | 2 | 3 | 4 | 5 | 6 | 7 | 8 | 9 | 10 | 11 |
|---|---|---|---|---|---|---|---|---|---|---|---|
| # Tasks | 5 | 16 | 29 | 26 | 14 | 18 | 7 | 1 | 0 | 3 | 1 |

Table 6: **Action space of Crab Benchmark-v0.** The actions at the top of the table apply to the Ubuntu environment, those in the middle to the Android environment, and those at the bottom are relevant across all environments.

| Action Name (Parameters) | Description |
|---|---|
| `click(elem)` | Click on `elem`. |
| `right_click(elem)` | Right-click on `elem`. |
| `double_click(elem)` | Double-click on `elem`. |
| `write_text(text)` | Typing the specified `text`. |
| `press(key)` | Press a keyboard `key`. |
| `hotkey(keys)` | Press keyboard `keys` at the same time. |
| `scroll(direction)` | Scrolls page up or down. |
| `search_app(name)` | Search for application with `name` in the system. |
| `tap(elem)` | Tap on `elem`. |
| `long_tap(elem)` | Press and hold `elem`. |
| `swipe(elem,dire,dist)` | Swipe from `elem` in a specified `direction` and `distance`. |
| `write_text(text)` | Typing the specified `text`. |
| `press(key)` | Press a `key`, can be *home* or *back*. |
| `show_all_drawer()` | Show the app drawer to list installed applications. |
| `submit(answer)` | Submit `answer` if needed. |
| `complete()` | State that a task is completed. |
| `wait()` | Wait the environment to process |

also as interfaces through which agents interact with the environment. The *evaluator* is a specialized *action* restricted to returning boolean values, signifying the success or failure of an agent's task. It enhances the *actions* by analyzing the state of the environment and the sequence of *actions* executed by the agent, providing a decisive metric of task accomplishment. Additionally, multiple *evaluators* can be interconnected to form a graph evaluator for complex tasks (Sec. 4.2).

The *benchmark* is a key definition in the framework. A benchmark includes multiple *environments* and cross-environment *tasks*. The *environment* is formed by an action space and an observation space, which are both defined by a list of *actions*, and other essential parameters necessary for its configuration. This composite structure facilitates the execution and monitoring of *actions*, whether on local machines, remote servers, virtual machines, or physical devices networked together. A *task* encapsulates a natural language description and a graph evaluator.

CRAB utilizes Python functions to define all actions and evaluators, embodying a "code as configuration" philosophy. Each function's docstring outlines its description and parameter definitions, which are then presented to the agent as structured prompts. Compared to traditional methods using data interchange formats like JSON or YAML, Python code configurations provide a more structured approach and fits in modern IDE.

By decoupling actions, environments, tasks, and evaluations, CRAB facilitates a plug-and-play architecture that can adapt to various scenarios. Such a system is scalable, maintainable and expandable, allowing researchers and developers to introduce new tasks and environments without restructuring the entire framework. Our implementation uses *networkx* (Hagberg et al.) for building graph and *dill* (McKerns et al.) for function serialization in our implementation.

A.4    CONFIGURATION BY MODULES

Building on the declarative and modular design of our framework, this section explains the configuration and potential extensibility of each module.

**Environment**   The environments in CRAB are a combination of multiple different uses of actions with some environment metadata, such as name and natural language description. In Crab Benchmark-v0, we use a computer desktop environment and a smartphone environment both based on virtual machine technology. The computer desktop environment, named *Ubuntu*, is installed from an ISO image of Ubuntu 22.04.4 LTS (Jammy Jellyfish) downloaded from the Ubuntu Official website[9]. Necessary applications such as the LibreOffice suite (Writer, Calc, and Impress) and Slack are installed later via snap and apt, according to the task dataset requirements. The smartphone environment, named *Android*, is installed using pre-defined devices (Google Pixel 8 Pro with release name *R*) provided in Google Android Studio[10]. We install additional required applications such as *Keep Notes*, *Tasks*, and *Docs* from Google Play. The descriptions of the two environments in Crab Benchmark-v0, which are inserted in the agent prompts, are as follows:

- **Ubuntu**: An Ubuntu 22.04 Linux desktop operating system. The interface displays a current screenshot at each step and primarily supports interaction via mouse and keyboard. You must use searching functionality to open any application in the system. This device includes system-related applications including Terminal, Files, Text Editor, Vim, and Settings. It also features Firefox as the web browser, and the LibreOffice suite—Writer, Calc, and Impress. For communication, Slack is available. The Google account is pre-logged in on Firefox, synchronized with the same account used in the Android environment.

- **Android**: A Google Pixel smartphone runs on the Android operating system. The interface displays a current screenshot at each step and primarily supports interaction through tapping and typing. This device offers a suite of standard applications including Phone, Photos, Camera, Chrome, and Calendar, among others. Access the app drawer to view all installed applications on the device. The Google account is pre-logged in, synchronized with the same account used in the Ubuntu environment.

**Action**   Action implementation in Crab Benchmark-v0 utilize the dynamic feature of Python. It provides an intuitive method to define actions through Python function. Here is an example of action `search_application` in the Ubuntu environment:

```python
@action
def search_application(name: str) -> None:
    """Search an application name.

    For exmaple, if you want to open an application named "slack",
    you can call search_application(name="slack"). You MUST use this
    action to search for applications.

    Args:
        name: the application name.
    """
    pyautogui.hotkey("win", "a")
    time.sleep(0.5)
    pyautogui.write(name)
    time.sleep(0.5)
```

Listing 1: Define "search_application" action.

We extract key information from the function through the `@action` decorator as following:

- **Name**: The action name serves as the identifier for backend models. It should semantically match the action's behavior to improve the accuracy of the agent in executing the action. The function name is extracted as the action name. In this example, `search_application` is the assigned name.

- **Description**: The description provides a natural language explanation of the action to assist the agent in understanding how to use it. The main body of the function's docstring is used as the description. For example, in this instance, the description outlines the basic usage of the action: *Search an application name*, along with an example of its usage.

---

[9]`https://releases.ubuntu.com/jammy/ubuntu-22.04.4-desktop-amd64.iso`
[10]`https://developer.android.com/studio`

- **Parameters**: The parameters are the arguments that the functions accept, offering flexibility for the agent to control the environment. Typically, a set of parameters is defined, each consisting of a name, type, and a natural language description. Parameters are extracted from the function's parameters along with their type annotations. Additionally, parameter descriptions are extracted from the `Args` section in the docstring. In this example, there is only one parameter named `name`, with a type of `str`, and its description is `the application name`.

- **Entry**: The entry represents the implementation of the function, defined within the function body to specify how the action is executed. When the agent invokes the function, the entry is executed with the provided parameters. In this example, we utilize the *pyautogui* package for keyboard control. Initially, it presses a hotkey to enter the application search panel in Ubuntu, then proceeds to type the application name provided by the parameters, finally displaying the search results.

**Observation**  The observation space is represented by a set of actions. These observation actions are designed to be parameter-free and return an observation result. For instance, within the Ubuntu environment, the sole observation action available is the `screenshot` function, defined as follows:

```python
@action
def screenshot() -> str:
    """Capture the current screen as a screenshot."""
    with mss() as sct:
    # Capture raw pixels from the screen
    sct_img = sct.grab(sct.monitors[1])
    # Convert to PNG format
    png = tools.to_png(sct_img.rgb, sct_img.size)
    # Encode to Base64 format for easier transmission
    base64_img = base64.b64encode(png).decode("utf-8")
    return base64_img
```

Listing 2: Define the "screenshot" observation action.

This action captures the screen's current view and encodes it in Base64 format. Additionally, visual prompts are also defined by actions that utilize the output from an observation action as their input, further processing it to generate a visual prompt for the agent.

**Evaluator**  The evaluator in Crab Benchmark-v0 is crafted to assess the outcome of actions performed by the agent within the environment. The evaluator is defined as an action that outputs a boolean value. An example of an evaluator in the Ubuntu environment is the `check_text_in_current_window_name` function, outlined below:

```python
@evaluator(env_name="ubuntu")
def check_text_in_current_window_name(text: str) -> bool:
    try:
        out = subprocess.check_output(
            ["xdotool", "getwindowfocus", "getwindowname"], text=True
        ).strip()
    except subprocess.CalledProcessError:
        return False
    return text in out
```

Listing 3: Define "check_text_in_current_window_name" evaluator.

The evaluator function is denoted with an `@evaluator` decorator and specifies its operating environment. The function's primary role is to execute a check within the system and return a boolean value indicating success or failure based on the condition being evaluated. Here, the function aims to verify whether a specified text appears in the title of the currently focused window. This is achieved through the use of the `subprocess` module to execute system commands that fetch the window's title, checking if the provided text parameter is contained within it.

**Task**  Following a declarative programming paradigm, the task is defined as a data model. Here is an example of a cross-platform task in the dataset:

```
Task(
    id="a3476778-e512-40ca-b1c0-d7aab0c7f18b",
    description="Open \"Tasks\" app on Android, check the...",
    evaluator=path_graph(
        check_current_package_name("com.google.android.apps.tasks"),
        check_current_window_process("gnome-control-center"),
        check_color_scheme("prefer-dark"),
    ),
)
```

Listing 4: Define a task.

In this model, each task is represented as an instance of the `Task` class, which is a subclass of `BaseModel` in *Pydantic*[11] package. Each task is uniquely identified by an ID and described by a detailed description. The evaluator component is structured as a graph evaluator, which integrates multiple evaluative functions into a directed graph using the *networkx*[12] package. Each evaluator within this graph must be appropriately parameterized to assess specific conditions relevant to the task. For example, the task demonstrated aims to open the "Tasks" app on Android and perform a series of verifications: it checks whether the correct Android app is opened, whether the current focused window's process name is `gnome-control-center`, and whether the color scheme is set to dark.

**Sub-task** The sub-task in CRAB is the unit component of in task construction. The following example is a sub-task template that we used to easily generate sub-tasks:

```
SubTask(
    id="0f589bf9-9b26-4581-8b78-2961b115ab49",
    description="Open \"{file_path}\" using vim in a terminal, write \"{
    content}\", then save and exit vim.",
    attribute_dict={"file_path": "file_path", "content": "message"},
    output_type="file_path",
    evaluator_generator=lambda file_path, content: path_graph(
        check_current_window_process("gnome-terminal-server"),
        is_process_open("vim"),
        is_process_close("vim"),
        check_file_content(file_path, content),
    ),
),
```

Listing 5: Define a task.

In this sub-task model, each sub-task is defined using a similar approach to the main task. The attributes of the sub-task are outlined in an `attribute_dict`, which details the types and roles of each attribute used in the sub-task's operations. The `output_type` field specifies the expected type of output from the sub-task. The types reflected in `attribute_dict` and `output_type`, play a critical role in determining the compatibility and sequential logic of compose multiple sub-tasks. The evaluator for the sub-task is dynamically generated using a lambda function, which crafts an evaluator sub-graph based on the sub-task's attributes.

A.5 COMPOSED TASK FORMAT

We use a JSON format to save the composed tasks, which includes the task ID, overall task description, sub-tasks with their attribute values, and a graph structure represented in an adjacency list. The entire task dataset is defined by the sub-task pool in Python code and the task composition JSON files categorized by task platform.

```
{
    "description": "Combine Image 1 \"/home/crab/Pictures/cat.png\" and
    Image 2 \"/home/crab/assets/campus.png\" using GIMP (GNU Image
```

---

[11]https://pydantic.dev/
[12]https://networkx.org/

```
   Manipulation Program), placing Image 1 on the left side of Image 2,
   and save the combined image to \"/home/crab/Desktop/background.png\".
    Then, set this combined image as the screen background of the system
   .",
   "tasks": [
       {
           "task": "4cf246ea-0a7f-43da-84b6-61d74a2699af",
           "attribute": {
               "image_path_1": "/home/crab/Pictures/cat.png",
               "image_path_2": "/home/crab/assets/campus.png",
               "output_path": "/home/crab/Desktop/background.png"
           },
           "output": "/home/crab/Desktop/background.png"
       },
       {
           "task": "a207ef38-b3b2-4c6c-a1e3-75c38162f5ba",
           "attribute": {
               "photo_path": "/home/crab/Desktop/background.png"
           },
           "output": null
       }
   ],
   "adjlist": "0 1\n1",
   "id": "d3c917ff-406f-447a-87f5-b8d835cba750"
}
```

Listing 6: Define a composite task in JSON.

## B    AGENT SYSTEM

### B.1    AGENT IMPLEMENTATION

In this section, we outline the implementation of the agents used in our experiments, which leverage advanced multimodal language models from OpenAI, Anthropic, and Google. Each agent is designed to function in multi-environment setups, interacting with various action spaces defined by different environments.

**General Framework**    All agents share a common architecture but are tailored to the specific APIs and capabilities of each language model provider.

**Initialization**    Each agent is initialized with several key parameters, including a description, an action space, the model type, maximum tokens, history message length, and an optional environment description. The initialization process involves:

- **Action Space Conversion**: Actions defined for each environment are converted into a schema compatible with the respective API. This ensures that the actions can be correctly interpreted and executed by the language models.
- **System Message Setup**: Depending on whether the agent is configured for single or multiple environments, a system message is formatted to provide the model with context about the tasks and environments.

**Interaction (Chat Method)**    The core functionality of each agent is encapsulated in its ability to interact with users through a chat method. This involves:

- **Content Parsing**: Input content is parsed and formatted to match the requirements of the respective API. This includes structuring user messages and any necessary contextual information.
- **Request Construction**: The request payload is constructed, incorporating the system message, chat history, and the newly parsed user input.

- **API Communication**: The constructed request is sent to the appropriate API, which generates a response. The agents handle API-specific constraints such as rate limits and response formats.

- **Response Handling**: The response from the API is processed to extract any tool calls suggested by the model. These are then appended to the chat history, maintaining a coherent conversation state.

**Multi-Environment Support** For agents configured to operate in multiple environments, additional logic ensures that actions are correctly associated with their respective environments. This involves modifying action names and descriptions to reflect their environmental context and handling responses accordingly.

**Utilities and Shared Functions** Several utility functions support the operation of these agents, facilitating tasks such as content parsing, action prompt generation, and schema conversion. These shared functions ensure consistency and reduce redundancy across the different agent implementations.

## B.2 INTER-AGENT COMMUNICATION STRATEGIES

In this section we introduce the details of two multi-agent communications methods, which are introduced in 6.1.

**Multi-agent Communication by Functionality** This setting involves two agents: a main agent prompted with the task description and a tool agent with the entire action space. The main agent generates the instruction for the next step and sends it to the tool agent. The tool agent chooses the proper action with parameters and a target environment, then feeds it back to the system.

**Multi-agent Communication by Environment** This setting involves four agents in our benchmark setting: a main agent prompted with the task description and three tool agents, each corresponding to the environments of Android, Ubuntu, and Root, with the respective action spaces. The main agent generates the instruction for the next step and sends it to the tool agents. Each sub-environment agent receives the message containing the instruction and environment observation information. The environment agents process the message using their specialized models and action schemas, performing the required actions within their environments.

## B.3 AGENT PROMPT

### B.3.1 SINGLE AGENT

> **Prompt**
>
> You are a helpful assistant. Now you have to do a task as described below:
> `**{task_description}**`.
> You should never forget this task and always perform actions to achieve this task. And this is the description of each given environment: `{env_description}`. A unit operation you can perform is called action in a given environment. For each environment, you are given a limited action space as function calls:
> `{action_descriptions}`
> You may receive a screenshot of the current system. You may receive a screenshot of a smartphone app. The interactive UI elements on the screenshot are labeled with numeric tags starting from 1.
> In each step, You MUST explain what do you see from the current observation and the plan of the next action, then use a provided action in each step to achieve the task. You should state what action to take and what the parameters should be. Your answer MUST be a least one function call. You SHOULD NEVER ask me to do anything for you. Always do them by yourself using function calls.

**Prompt**

You are a helpful assistant. Now you have to do a task as described below:
`**{task_description}**`
You should never forget this task and always perform actions to achieve this task. And this is the description of each given environment: `{env_description}`. You will receive screenshots of the environments. The interactive UI elements on the screenshot are labeled with numeric tags starting from 1.
A unit operation you can perform is called Action. You have a limited action space as function calls: `{action_descriptions}`. You should generate JSON code blocks to execute the actions. Each code block MUST contains only one json object, i.e. one action. You can output multiple code blocks to execute multiple actions in a single step. You must follow the JSON format below to output the action.
`{"name": "action_name", "arguments": {"arg1": "value1",`
`"arg2": "value2"}}`
or if not arguments needed:
`{"name": "action_name", "arguments": {}}`
You MUST use exactly the same "action_name" as I gave to you in the action space. You SHOULDN'T add any comments in the code blocks.
In each step, You MUST explain what do you see from the current observation and the plan of the next action, then use a provided action in each step to achieve the task. You should state what action to take and what the parameters should be. Your answer MUST contain at least one code block. You SHOULD NEVER ask me to do anything for you. Always do them by yourself.

### B.3.2 MULTI-AGENT BY FUNCTIONALITY

**Main Agent Prompt**

You are a helpful assistant. Now you have to do a task as described below: `{task_description}`. And this is the description of each given environment: `{env_description}`. A unit operation you can perform is called action in a given environment. For each environment, you are given a limited action space as function calls: `{action_descriptions}`
You may receive a screenshot of the current system. The interactive UI elements on the screenshot are labeled with numeric tags starting from 1. For each step, You must state what actions to take, what the parameters are, and you MUST provide in which environment to perform these actions.

**Tool Agent Prompt**

You are a helpful assistant in generating function calls. I will give you a detailed description of what actions to take next, you should translate it into function calls. please do not output any other information.

### B.3.3 MULTI-AGENT BY ENVIRONMENT

**Main Agent Prompt**

You are a main agent, and your goal is to plan and give instructions to sub-agents in each environment to complete the final task. Now you have to do a task as described below: `{description}`. The description of each given environment: `{env_description}`. For each step, you are required to provide high-level instructions detailing the next actions to be taken. Additionally, you must specify which sub-agent in the designated environment should execute these instructions. If a sub-agent is not needed for a particular step, you may instruct it to skip that step.

> **Root Environment Agent Prompt**
>
> You are a sub-agent responsible for the crab benchmark root environment. Your goal is to assist the main agent in completing the whole task: "`{description}`". You can only complete the task or submit the result when the main agent tells you the whole task has been completed. Otherwise, you can only call SKIP.

> **Sub-environment Agent Prompt**
>
> You are a sub-agent responsible for the `{environment}` environment. The description of the `{environment}` environment is: `{env_description}`. Your goal is to assist the main agent in completing the final task by performing actions in the `{environment}` environment according to the instructions from the main agent. The final task is described below: `{task_description}`. A unit operation you can perform is called action in a given environment. You can only execute action in the `{environment}` environment. For the `{environment}` environment, you are given a limited action space as function calls: `{action_descriptions}`
> The interactive UI elements on the screenshot are labeled with numeric tags starting from 1. For each step, You will receive an instruction telling you what you need to do next. After analyzing the instruction you received and the current `{environment}` system, if you think you don't need to do anything in the current `{environment}` system, you should choose SKIP action. Otherwise, you must state what actions to take, what the parameters are, and you MUST provide in which environment to perform these actions. Your answer must be function calls. Please do not output any other information. You must make sure all function calls get their required parameters.

## C  FURTHER RESULT ANALYSIS

This section further discusses our experimental results in detail. Section C.1 categorizes the results into three types of tasks: Ubuntu, Android, and cross-platform, and provides further analysis. Section C.3 examines three specific tasks and analyzes the performance of different agent settings on each.

### C.1  RESULT BY PLATFORMS

Table 7, 8 and 9 show the experiment results on Ubuntu Tasks, Android Tasks, and cross-platform Tasks, respectively.

We find that certain models demonstrate a distinct preference or better alignment with specific platforms. The GPT-4o, Gemini, and Claude models, for instance, show notably better outcomes on Android platforms. This suggests potential optimizations or intrinsic features within these models that cater effectively to the Android environment's requirements. Conversely, the GPT-4 Turbo model exhibits superior performance on Ubuntu tasks, hinting at possible architectural or training aspects that are better suited for that specific environment.

Cross-platform tasks necessitate functionality across different operating systems or platforms, demand a broader capability range and more sophisticated agent coordination. The importance of CR is especially critical in such environments, where it serves as a more reliable metric for distinguishing between agent models than SR. Given the presence of all Gemini, Claude, and open source model agents' SR is 0.0, indicating that Completion Ratio more effectively captures an agent model's capability, thereby better reflecting its robustness and adaptability to complex requirements. On cross-platform tasks, GPT-4 Turbo (Single) exhibits a CR of 52.61%, which indicates that even though SR might be lower, the agent covers a significant portion of task objectives before termination.

Furthermore, analyzing the reasons for task termination offers additional insights into the operational challenges these models encounter. False Completion is notably prevalent in Android tasks. Reach Step Limit remains the most frequent cause of termination, particularly in cross-platform tasks. The Claude model exhibits a significantly high Invalid Action ratio in cross-platform tasks, indicating its difficulties in managing multi-environment scenarios effectively. The GPT-4o with JSON mode

Table 7: **Evaluation results on Ubuntu tasks.**

| Agent system | | Metrics | | | | Termination Reason | | |
|---|---|---|---|---|---|---|---|---|
| Model | Structure | SR(%) ↑ | CR(%) ↑ | EE(%) ↑ | CE(%) ↑ | FC(%) | RSL(%) | IA(%) |
| GPT-4O | Single | 9.59 | 30.82 | 3.22 | $4.87 \times 10^{-4}$ | 6.85 | 58.90 | 24.66 |
| GPT-4O | By Func | 9.59 | 24.20 | 2.72 | $4.30 \times 10^{-4}$ | 5.48 | 63.01 | 21.92 |
| GPT-4O | By Env | 10.96 | 22.88 | 2.74 | $2.29 \times 10^{-4}$ | 5.48 | 43.84 | 39.73 |
| GPT-4 TURBO | Single | 10.96 | **31.09** | **4.08** | **5.57** $\times 10^{-4}$ | 2.74 | 65.75 | 20.55 |
| GPT-4 TURBO | By Func | **12.33** | 28.95 | 3.70 | $4.18 \times 10^{-4}$ | 8.22 | 32.88 | 46.58 |
| GEMINI 1.5 PRO | Single | 1.37 | 7.76 | 0.63 | n/a | 0.00 | 47.95 | 50.68 |
| GEMINI 1.5 PRO | By Func | 1.37 | 3.31 | 0.33 | n/a | 0.00 | 20.55 | 78.08 |
| CLAUDE 3 OPUS | Single | 0.00 | 9.54 | 0.72 | $0.63 \times 10^{-4}$ | 8.22 | 58.90 | 32.88 |
| CLAUDE 3 OPUS | By Func | 0.00 | 4.93 | 0.46 | $0.47 \times 10^{-4}$ | 27.40 | 34.25 | 38.36 |
| GPT-4O W/O FC | Single | 10.96 | 22.58 | 2.30 | $4.49 \times 10^{-4}$ | 5.48 | 54.79 | 28.77 |
| PIXTRAL-12B | Single | 0.00 | 2.97 | 0.22 | $0.24 \times 10^{-4}$ | 1.37 | 80.82 | 17.81 |
| LLAVA-OV-72B | Single | 0.00 | 3.31 | 0.20 | $0.35 \times 10^{-4}$ | 17.81 | 64.38 | 17.81 |

shows a extremely high IA ratio in Android tasks, proving the serious hallucination problem under this setting.

Overall, these findings underscore the necessity of selecting the appropriate agent model and configuration based on specific platform and task needs. The variability in model performance across different setups also highlights the ongoing need for development and refinement of multi-agent systems to enhance their versatility and efficacy in increasingly diverse and complex operational environments. These results comparing SR and CR also demonstrates the important of our graph evaluator in agent evaluation.

## C.2 COMPARISON BETWEEN SINGLE AGENT AND MULTI-AGENT

The experimental results indicate that multi-agent structures perform slightly worse than single-agent systems, which is somewhat unusual. We analyse the possible reasons here.

First, comparing in False Completion Rate, we attribute the lower Success Rate (SR) of Multi-agent to a high False Completion Rate—where the agent incorrectly assumes that the task is complete. As observed in failure cases (e.g., the Cross-platform Task case study in Appendix C.3), Sub-agents often misinterpret the Main agent's instructions. Despite being required to perform a final action, the instructions lead Sub-agents to prematurely conclude that the task is complete, resulting in incorrect "complete" actions. While this issue also occurs in Multi-Env, it happens less frequently. By analysing the communication logs, we believe this is due to information loss during inter-agent communication. Sometimes, the main agent gives a correct instruction, but the sub-agent misunderstands it because it does not have the context. Natural language, while effective for aligning with human understanding in LLM communication, is less suited for inter-agent communication, leading to information loss during compression and interpretation, which weakens the performance of multi-agent structures.

Next, comparing in Invalid Action Rate, we observe that in single-platform tasks, both Multi-Env and Multi-Func suffer from similar inter-agent communication issues, as indicated by their high Invalid Action rates. However, in cross-platform tasks (Table 9), the Single agent's Invalid Action rate is significantly higher than that of the Multi-agent by environment structures on GPT-4o model. Cross-platform tasks require frequent environment changes with varying action spaces, and if the model's performance output is inadequate, it often generates correct actions in the wrong environment, invalid actions in the correct environment, or correct actions in correct environment but in the wrong format. This phenomenon highlights the limitations of current general-purpose LLMs, where multi-agent structures can be advantageous. By assigning each agent a specific responsibility and a limited action space, multi-agent structures can mitigate these issues.

To improve multi-agent system performance, we suggest to follow two approaches: (1) Developing better multi-agent structures to minimize information loss during communication, and (2) Introducing a critical agent to correct hallucinations or information loss during communication. These

Table 8: **Evaluation results on Android tasks.**

| Agent system | | Metrics | | | | Termination Reason | | |
|---|---|---|---|---|---|---|---|---|
| Model | Structure | SR(%) ↑ | CR(%) ↑ | EE(%) ↑ | CE(%) ↑ | FC(%) | RSL(%) | IA(%) |
| GPT-4O | Single | 24.14 | 47.91 | 5.84 | $7.17 \times 10^{-4}$ | 13.79 | 58.62 | 3.45 |
| GPT-4O | By Func | 24.14 | 48.74 | 6.83 | **9.19** $\times 10^{-4}$ | 24.14 | 37.93 | 13.79 |
| GPT-4O | By Env | **27.59** | **53.34** | **6.99** | $4.58 \times 10^{-4}$ | 13.79 | 44.83 | 13.79 |
| GPT-4 TURBO | Single | 6.90 | 27.08 | 2.60 | $2.87 \times 10^{-4}$ | 20.69 | 62.07 | 10.34 |
| GPT-4 TURBO | By Func | 20.69 | 37.01 | 5.00 | $5.92 \times 10^{-4}$ | 13.79 | 51.72 | 13.79 |
| GEMINI 1.5 PRO | Single | 17.24 | 34.52 | 4.82 | n/a | 10.34 | 65.52 | 6.90 |
| GEMINI 1.5 PRO | By Func | 17.24 | 35.99 | 4.31 | n/a | 31.03 | 37.93 | 13.79 |
| CLAUDE 3 OPUS | Single | 13.79 | 41.90 | 5.07 | $5.37 \times 10^{-4}$ | 20.69 | 55.17 | 10.34 |
| CLAUDE 3 OPUS | By Func | 13.79 | 44.02 | 4.75 | $5.35 \times 10^{-4}$ | 48.28 | 31.03 | 6.90 |
| GPT-4O w/o FC | Single | 10.34 | 14.29 | 1.72 | $2.94 \times 10^{-4}$ | 3.45 | 6.90 | 79.31 |
| PIXTRAL-12B | Single | 3.45 | 24.17 | 2.16 | $2.72 \times 10^{-4}$ | 0.00 | 65.52 | 31.03 |
| LLAVA-OV-72B | Single | 3.45 | 13.51 | 1.36 | $3.00 \times 10^{-4}$ | 3.45 | 93.10 | 0.00 |

Table 9: **Evaluation results on cross-platform tasks.**

| Agent system | | Metrics | | | | Termination Reason | | |
|---|---|---|---|---|---|---|---|---|
| Model | Structure | SR(%) ↑ | CR(%) ↑ | EE(%) ↑ | CE(%) ↑ | FC(%) | RSL(%) | IA(%) |
| GPT-4O | Single | 16.67 | 51.24 | **5.21** | **3.98** $\times 10^{-4}$ | 5.56 | 38.89 | 38.89 |
| GPT-4O | By Func | **22.22** | 50.00 | 4.15 | $3.13 \times 10^{-4}$ | 11.11 | 44.44 | 22.22 |
| GPT-4O | By Env | 5.56 | 43.54 | 3.22 | $1.60 \times 10^{-4}$ | 11.11 | 72.22 | 11.11 |
| GPT-4 TURBO | Single | 5.56 | **52.61** | 4.60 | $2.89 \times 10^{-4}$ | 11.11 | 66.67 | 16.67 |
| GPT-4 TURBO | By Func | 5.56 | 46.17 | 4.06 | $2.67 \times 10^{-4}$ | 16.67 | 50.00 | 27.78 |
| GEMINI 1.5 PRO | Single | 0.00 | 16.14 | 1.15 | n/a | 0.00 | 72.22 | 27.78 |
| GEMINI 1.5 PRO | By Func | 0.00 | 13.65 | 1.21 | n/a | 5.56 | 77.78 | 16.67 |
| CLAUDE 3 OPUS | Single | 0.00 | 24.50 | 1.93 | $1.24 \times 10^{-4}$ | 0.00 | 55.56 | 44.44 |
| CLAUDE 3 OPUS | By Func | 0.00 | 18.96 | 1.93 | $1.20 \times 10^{-4}$ | 0.00 | 38.89 | 61.11 |
| GPT-4O w/o FC | Single | 0.00 | 39.11 | 3.51 | $3.28 \times 10^{-4}$ | 5.56 | 50.00 | 44.44 |
| PIXTRAL-12B | Single | 0.00 | 12.35 | 0.62 | $0.44 \times 10^{-4}$ | 0.00 | 72.22 | 27.78 |
| LLAVA-OV-72B | Single | 0.00 | 9.07 | 0.48 | $0.53 \times 10^{-4}$ | 5.56 | 66.67 | 27.78 |

improvements, however, come with a trade-off, namely an increase in token costs within the agent system. Within our benchmark framework, users can utilize the error log we provide to analyze the bottlenecks of their agents and refine their designs.

## C.3 CASE STUDY

To better understand how different agents perform the same task and exhibit varied properties, we present visual results along with detailed metrics and logs for three cases by platform. The screenshots illustrate the progress of agents executing tasks according to specific natural language instructions.

### C.3.1 CROSS-PLATFORM TASK

**Task: Open the "Tasks" app on an Android device, check the first incomplete task, and then execute it as described.** The first task, found incomplete in the "Tasks" app, involves **switching the system to dark mode in Ubuntu via the "Settings" application.**

This task exemplifies message passing across different environments, where the "incomplete task" serves as the critical information that the agent must relay and apply in the Ubuntu setting. These two phases—retrieving the task details via the phone and executing the task on a computer—are inseparably linked and cannot be treated as distinct tasks. The agent can only proceed to the second stage after successfully acquiring information from the first.

In this task, GPT-4o (single agent), GPT-4 Turbo (single agent), and GPT-4 Turbo (multi-agent by functionality) all successfully complete the task using the minimal steps necessary to locate and execute the task, demonstrating their efficiency in managing multiple environments simultaneously. On the other hand, both GPT-4o (multi-agent by functionality) and GPT-4o (multi-agent by environment) also perform commendably, completing the task up until the final step. However, after incorrectly performing the last step, they both erroneously conclude the task is completed and exit. This indicates a communication breakdown, where the sub-agents misinterpret the instructions from the main agent. The remaining four agents fail to complete the task. Agents equipped with the Gemini model do not even manage to open the "Tasks" app within the allocated step limit, whereas agents with the Claude model quickly open the "Tasks" app to complete the first step but fail at the task execution. The performance disparity between single-agent and multi-agent configurations in both the Gemini and Claude models highlights the variance in capability across different models and devices.

### C.3.2 UBUNTU TASK

**Task: Create a new directory "/home/crab/assets_copy" and copy all files with the specified "txt" extension from "/home/crab/assets" to the directory "/home/crab/assets_copy".**

This task can be approached through multiple methods. An agent may opt for a straightforward strategy first using the `search_application` command to find the Terminal, then using Linux commands to create the directory and copy the necessary files. Alternatively, the agent could employ a GUI-based approach, manually creating the folder and selecting files through actions like `click` and `right_click`. We evaluate various agent systems in a single-agent setting for this task. As illustrated in Table 10–13 , both GPT-4o and GPT-4 Turbo from OpenAI successfully interpret the task instructions and employ a simpler solution using Terminal commands. These agents also demonstrate superior capability in understanding the UI, selecting the correct commands, and accurately using the Terminal application to fulfill the task requirements.

Conversely, the Gemini and Claude agents, despite attempting to solve the task with Terminal, ultimately fail in different ways. Both agents struggle with precise clicking and selecting the correct icons for the intended actions, even though they share the same visual prompting mechanism as GPT-4o and GPT-4 Turbo. For instance, the Claude agent mistakenly opens the Ubuntu Desktop Guide instead of the Terminal and continues executing commands in the wrong application without realizing the error. The Gemini agent, on the other hand, unexpectedly opens the Firefox browser before correctly navigating to the Terminal but still interacts incorrectly with unrelated applications and icons. Unlike Claude, Gemini does not type in commands in the wrong applications but persists in exploring alternative methods using the Files application's UI. Despite taking significantly more steps than the GPT-4o and GPT-4 Turbo agents, neither the Claude nor the Gemini agents achieve the task's goal.

### C.3.3 ANDROID TASK

**Task: In Android, using the "Contacts" app, find the email of the contact named John Lauphin, then using the "Gmail" app, send an email to that contact with the subject "Hello John."**

This task consists of sub-tasks across two different applications. Agents must sequentially open the two apps, retrieve the email address from the first app, and use it in the second app to send an email. This straightforward yet formal task can be completed using various methods. Agents may need to locate the contact in the Contacts app and then use the retrieved email address to send a message. We reports the performance of agents in a multi-agent setting for this challenging task. Following is the details of agents in operating the task.

**GPT-4o multi-agent by functionality** In steps 1-11, the agent tries to open the Contacts app but mistakenly opens Google Assistant multiple times. In steps 12-14, the agent successfully enters the Contacts app and finds the contact information. The agent then returns to the home page, and the process is terminated due to the limitation of operation steps.

**GPT-4 Turbo multi-agent by functionality** In steps 1-2, the agent tries to open the Contacts app but mistakenly opens Google Messages. In steps 3-5, the agent opens the Contacts app and obtains the

corresponding information. In steps 6-14, the agent repeatedly opens Google Chrome and Messages apps, failing to find the Gmail app as planned.

**Gemini 1.5 Pro multi-agent by functionality** In steps 1-2, the agent finds the Contacts app and enters it. However, the agent misunderstands the instruction, gets lost in creating a new contact with the given name, and cannot obtain the corresponding information.

**Claude 3 Opus multi-agent by functionality** In steps 1-7, the agent tries to open the Contacts app but mistakenly opens Google Messages multiple times. In steps 7-11, the agent tries to open the Contacts app but mistakenly opens Google Assistant. In steps 12-14, the agent successfully enters the Contacts app and finds the contact information. The agent then returns to the home page, plans to open the Gmail app, and the process is terminated due to the limitation of operation steps.

**GPT-4o multi-agent by environment** In steps 1-7, the agent plans to open the Contacts app, but the operation fails due to an error in opening the app drawer, which prevents the agent from finding and tapping the Contacts app. In steps 8-11, the agent successfully enters the Contacts app and obtains the information. In steps 12-14, the agent opens the Gmail app, navigates to the sending page, and tries to input the retrieved email address as the recipient.

**Analysis** For the agents which are organized by functionality, Gemini 1.5 Pro struggles to complete the first operation. Although it recognizes and opens the Contacts app as instructed, it fails to proceed further. In contrast, Claude 3 Opus and GPT-4o successfully obtain the necessary information. In the initial phase, the multiple agents agree that opening the Contacts app is the first step. However, they often fail to find the correct position to tap, frequently opening incorrect apps such as Google Assistant and Messages. Once the agents do open the correct app, they usually find the email address of the contact quickly. Even when agents plan to go back home and open the Gmail app to send the message, due to the limitation of operations, the system ended. As shown in steps 3-5, GPT-4 Turbo quickly finishes the corresponding task after opening the correct app. However, similar to GPT-4o, GPT4-Turbo agents get stuck as they can not open the correct apps in the following steps. Besides, GPT-4o (multi-agent by environment) overcomes the issue encountered by GPT-4o (multi-agent by functionality). Even affected by not being able to access the app drawer, the system could still find and copy the corresponding information and change to the Gmail app for further operations.

Table 10: **Ubuntu task case with Gemini (Single):** Create a new directory "/home/crab/assets_copy" and copy all files with the specified "txt" extension from "/home/crab/assets" to the directory "/home/crab/assets_copy".

| Step | Agent Observation and Action |
|------|------------------------------|
| 0 | 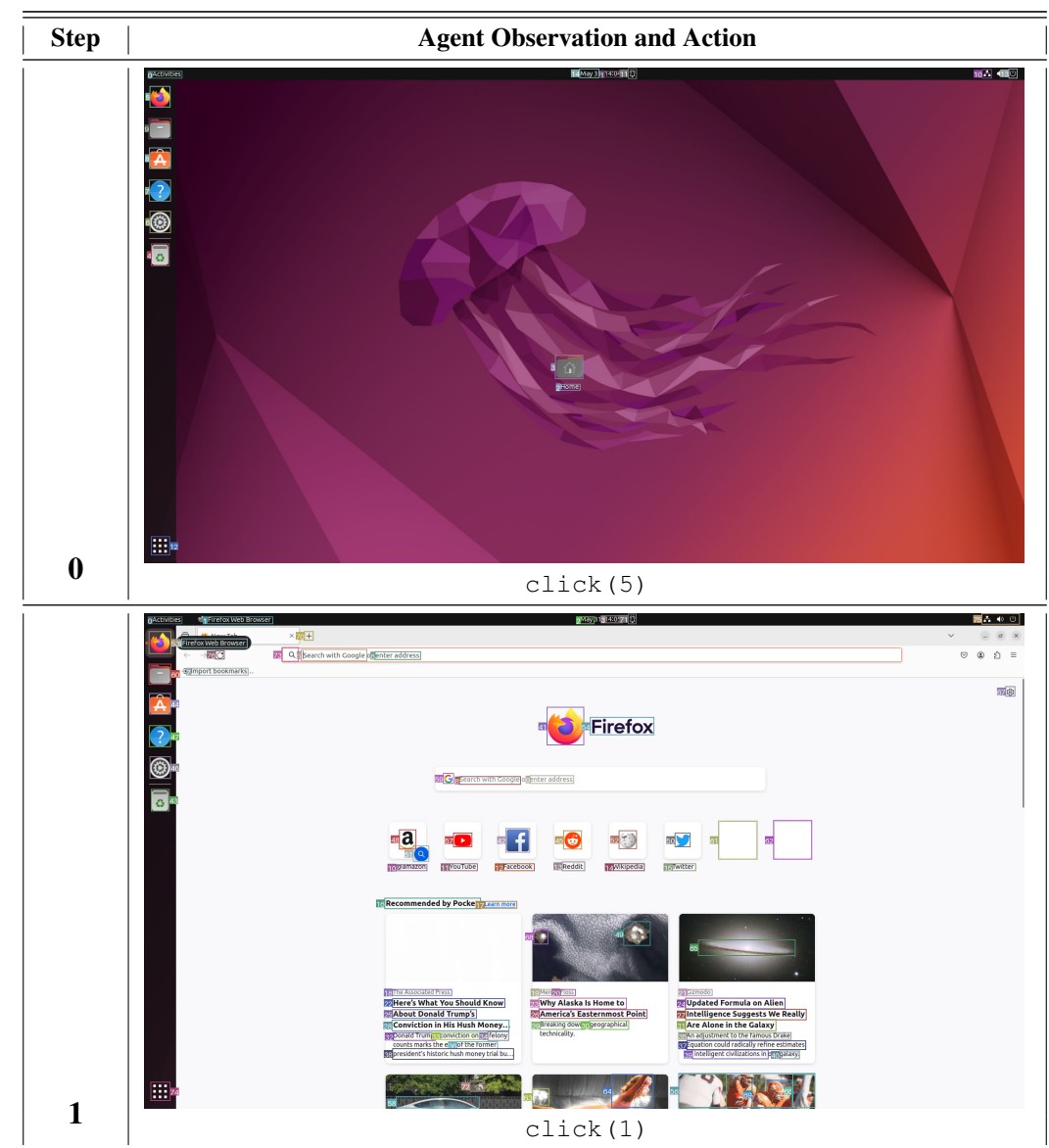 `click(5)` |
| 1 | `click(1)` |

**Table 10 continued from previous page**

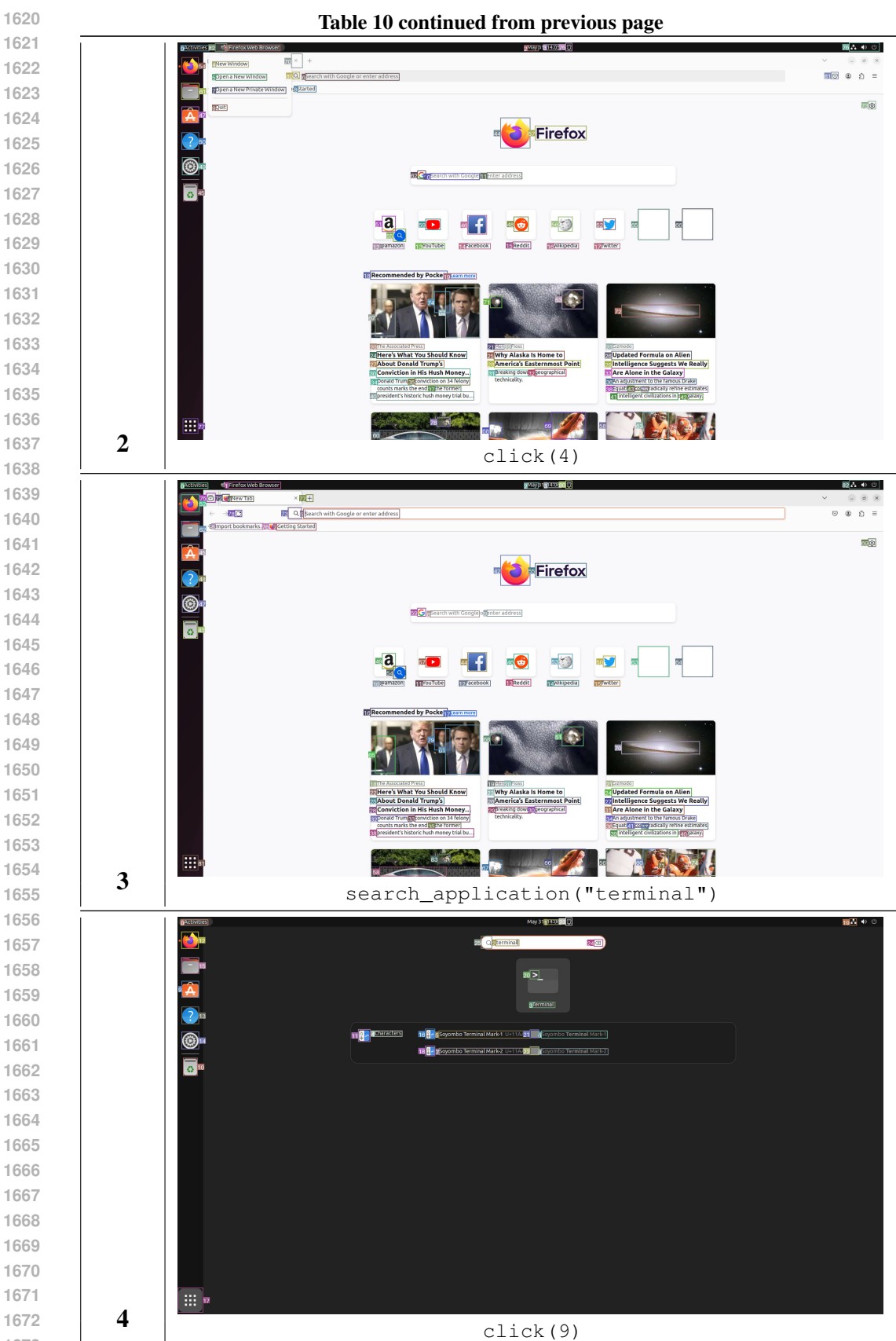

| | |
|---|---|
| 2 | click(4) |
| 3 | search_application("terminal") |
| 4 | click(9) |

**Table 10 continued from previous page**

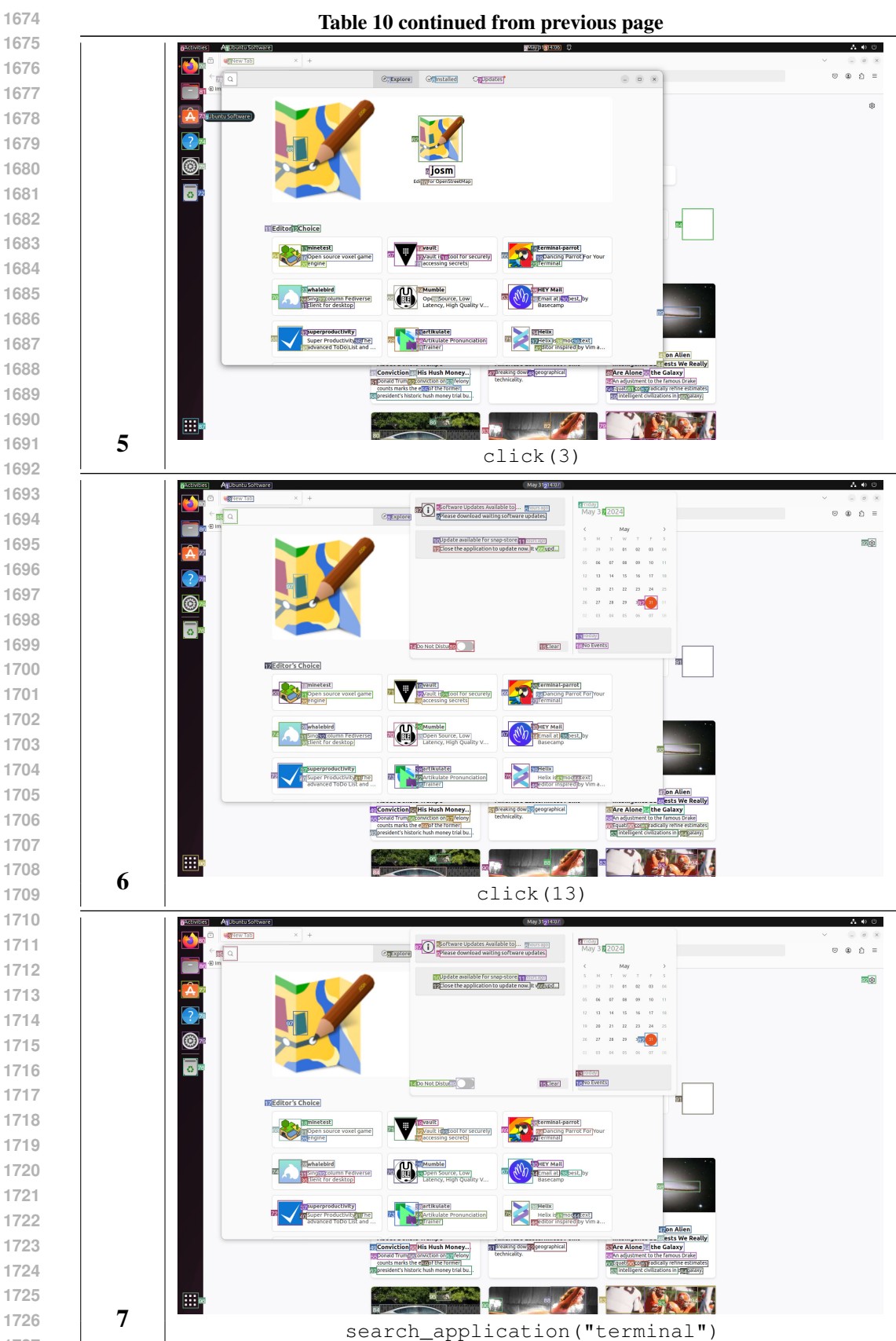

5     `click(3)`

6     `click(13)`

7     `search_application("terminal")`

**Table 10 continued from previous page**

8 | `click(10)`

9 | `click(6)`

10-14 | The agent is stuck at this stage and keeps clicking useless elements.

**Completed Nodes / Total Nodes**: 0 / 2   **Termination Reason**: Reach Step Limit

Table 11: **Ubuntu task case with Claude (Single):** Create a new directory "/home/crab/assets_copy" and copy all files with the specified "txt" extension from "/home/crab/assets" to the directory "/home/crab/assets_copy".

| Step | Agent Observation and Action |
|---|---|
| 0 | 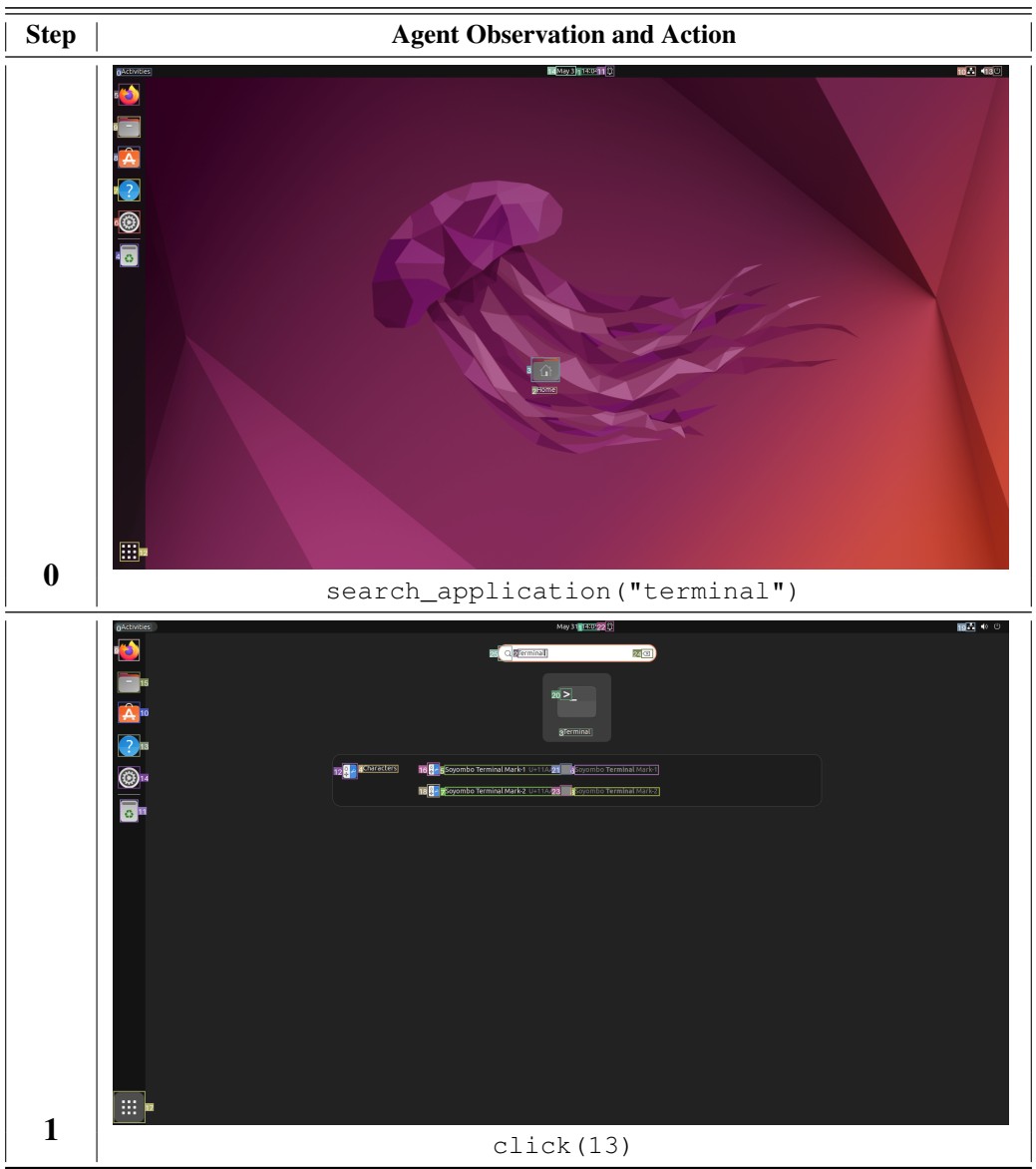 search_application("terminal") |
| 1 | click(13) |

Continued on next page

**Table 11 continued from previous page**

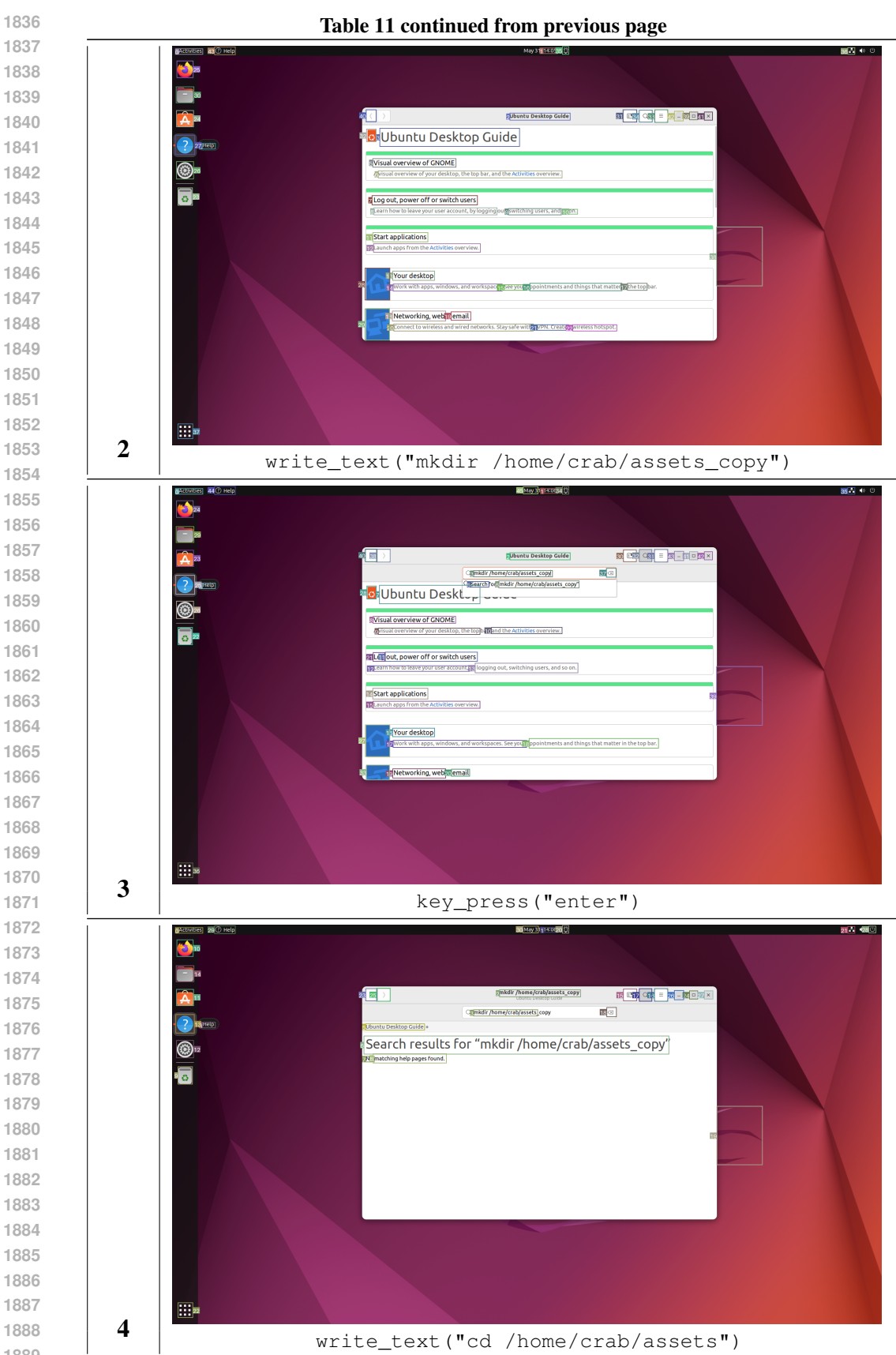

| | |
|---|---|
| 2 | `write_text("mkdir /home/crab/assets_copy")` |
| 3 | `key_press("enter")` |
| 4 | `write_text("cd /home/crab/assets")` |

**Table 11 continued from previous page**

| | |
|---|---|
| **5** |  `key_press("enter")` |
| **6** |  `write_text("mkdir assets_copy")` |
| **7-14** | The agent is stuck at this stage and keeps pressing keys. |

**Completed Nodes / Total Nodes**: 0 / 2   **Termination Reason**: Reach Step Limit

Table 12: **Ubuntu task case with GPT-4o (Single):** Create a new directory "/home/crab/assets_copy" and copy all files with the specified "txt" extension from "/home/crab/assets" to the directory "/home/crab/assets_copy".

| Step | Agent Observation and Action |
|---|---|
| 0 | 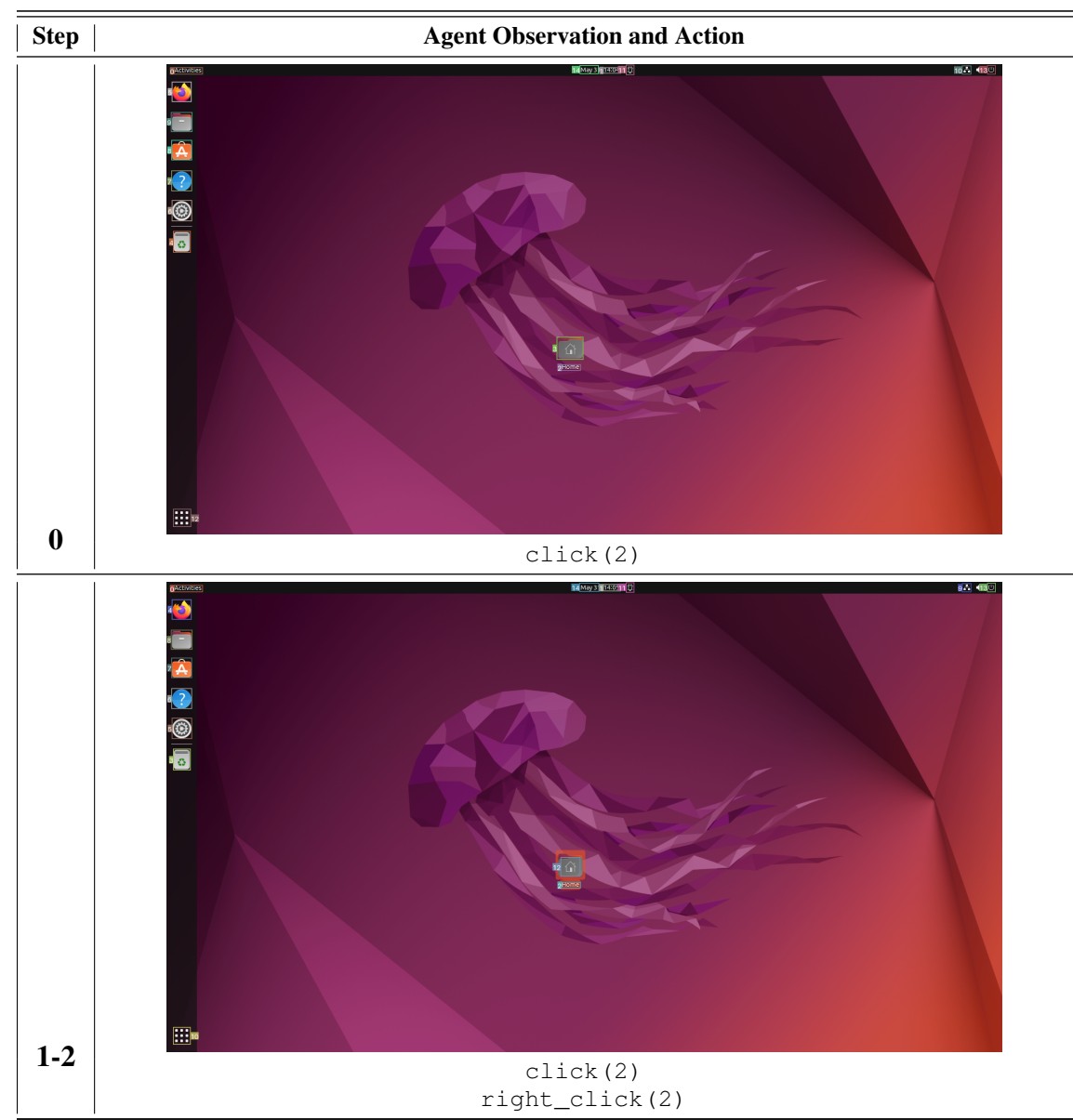
click(2) |
| 1-2 | click(2)
right_click(2) |

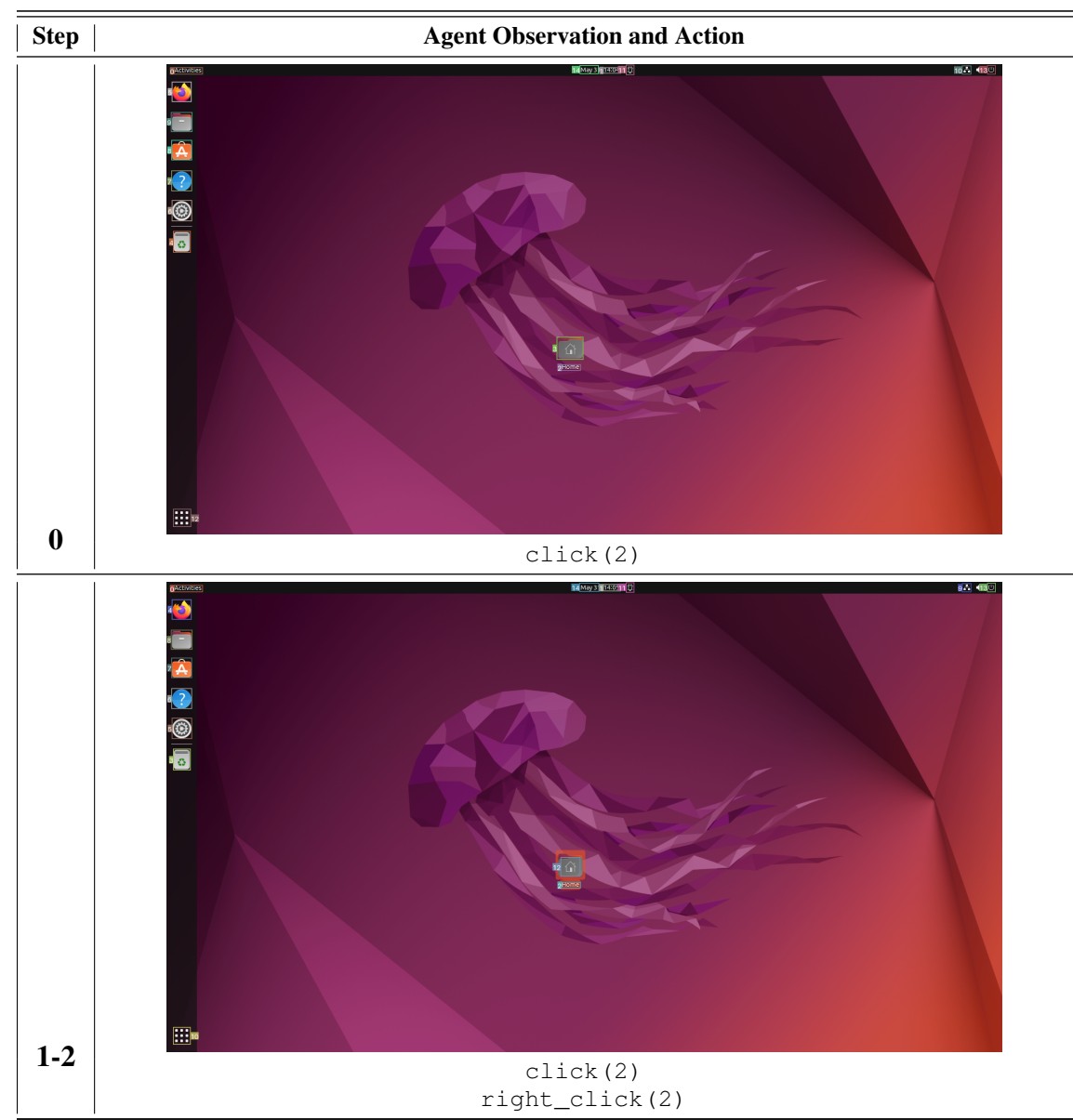

**Table 12 continued from previous page**

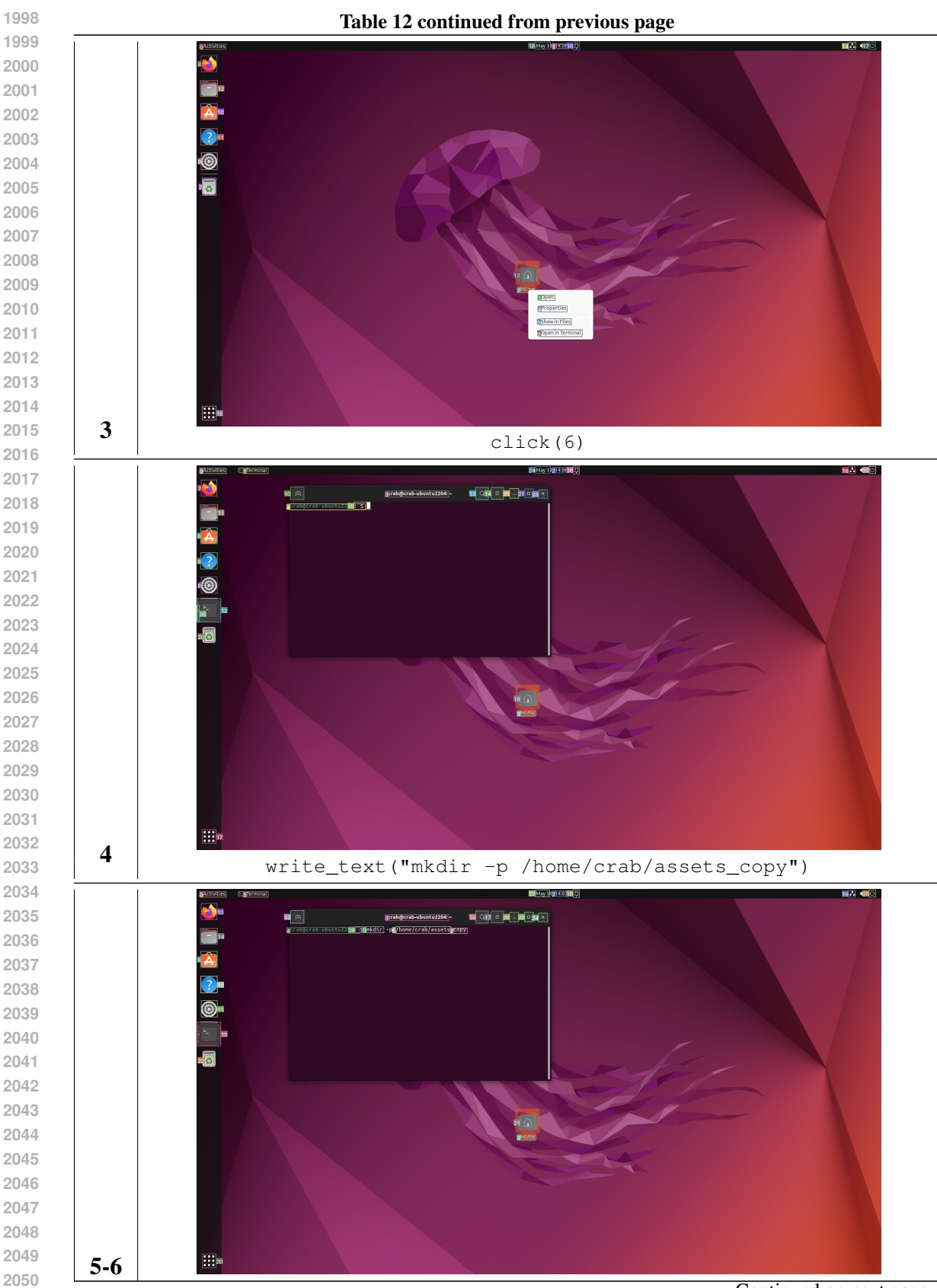

| | |
|---|---|
| 3 | click(6) |
| 4 | write_text("mkdir -p /home/crab/assets_copy") |
| 5-6 | |

**Table 12 continued from previous page**

| 5-6 | `key_press("enter")`
`write_text("cp /home/crab/assets/*.txt /home/crab/assets_copy/")` |
|---|---|
| 7 | 
`key_press("enter")` |

**Completed Nodes / Total Nodes**: 2 / 2   **Termination Reason**: Success

Table 13: **Ubuntu task case with GPT-4 Turbo (Single):** Create a new directory "/home/crab/assets_copy" and copy all files with the specified "txt" extension from "/home/crab/assets" to the directory "/home/crab/assets_copy".

| Step | Agent Observation and Action |
|---|---|
| 0 | 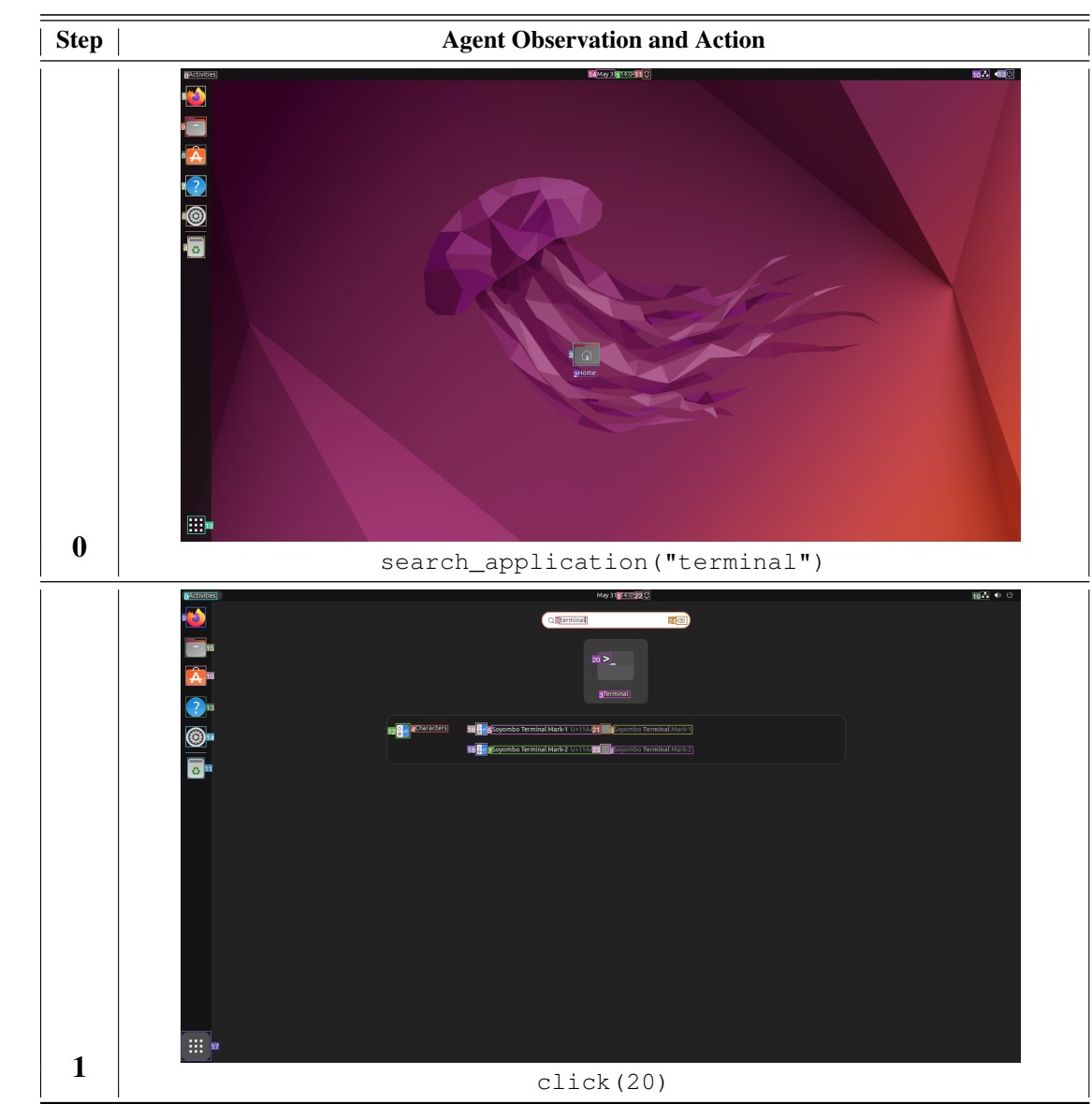
search_application("terminal") |
| 1 | click(20) |

**Table 13 continued from previous page**

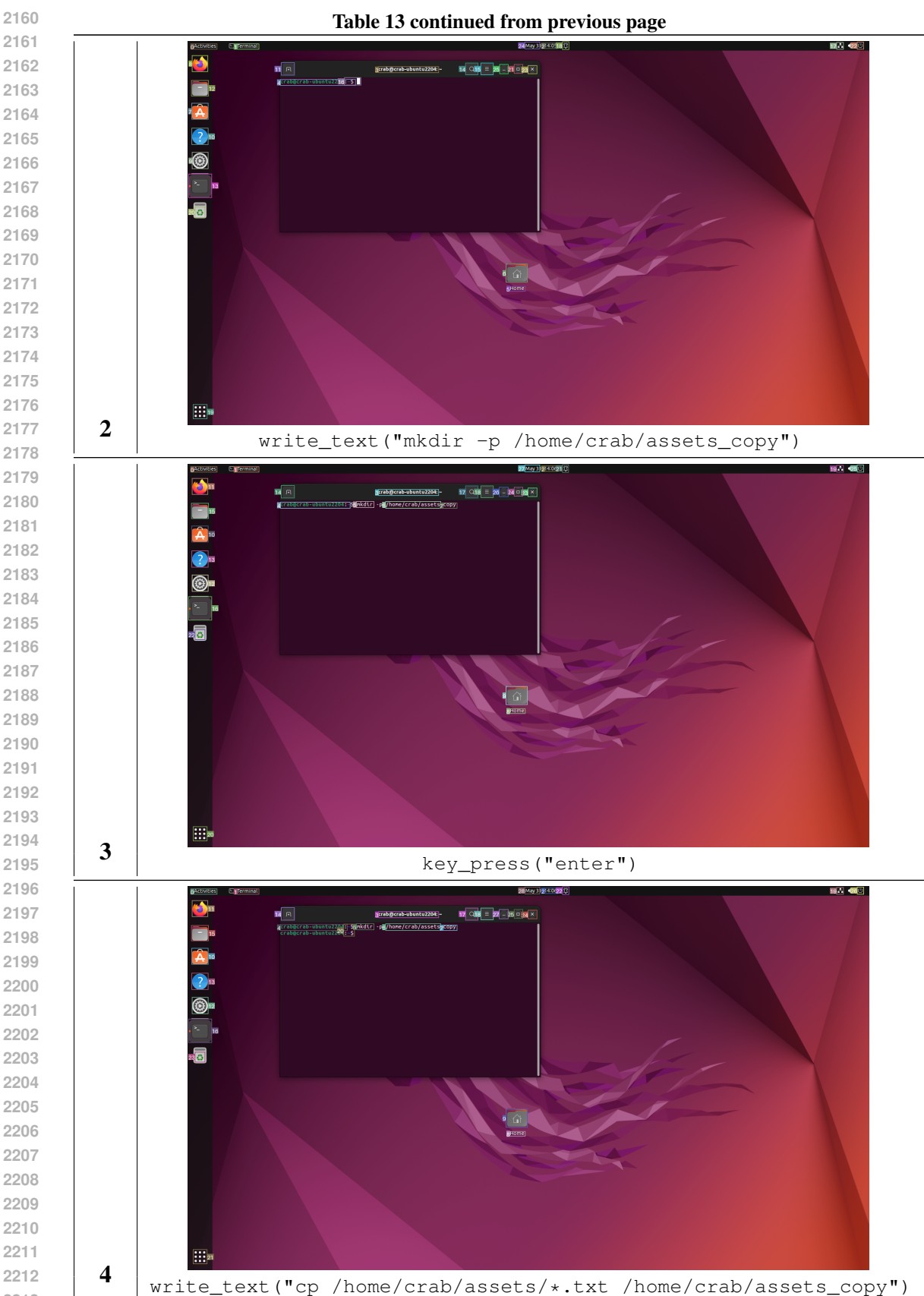

| 2 | write_text("mkdir -p /home/crab/assets_copy") |

| 3 | key_press("enter") |

| 4 | write_text("cp /home/crab/assets/*.txt /home/crab/assets_copy") |

**Table 13 continued from previous page**

| 5 |  |
| --- | --- |
| | `key_press("enter")` |

**Completed Nodes / Total Nodes**: 2 / 2   **Termination Reason**: Success

