# OpenReview forum: "CRAB: Cross-environment Agent Benchmark for Multimodal Language Model Agents"
_ICLR.cc/2025/Conference — Submitted to ICLR 2025_

### Official Review · Reviewer_qvQK · 2024-11-02

**Soundness:** 2
**Presentation:** 2
**Contribution:** 3
**Rating:** 5
**Confidence:** 3

**Summary:**

This paper introduces CRAB, a new framework for evaluating multimodal language model (MLM) agents across different computing environments like desktop and mobile platforms. The key contributions include: (1) The first benchmark framework supporting cross-environment tasks, (2) A graph-based evaluation method for fine-grained assessment of agent performance, and (3) An efficient mechanism for constructing tasks and evaluators. The authors implement Crab Benchmark-v0 with 120 tasks across Ubuntu desktop and Android environments, evaluating several MLMs in different agent configurations. The best performance achieved was 38.01% completion ratio using GPT-4o in a single-agent setup.

**Strengths:**

1. The paper considers an important problem in MLM agent evaluation: cross-environment evaluation. Evaluating MLM agents across different platforms better reflects real-world scenarios where tasks often require coordination between devices.
2. The benchmark demonstrates good diversity in its evaluation coverage, which helps provide a more comprehensive assessment of MLM agents' capabilities.

**Weaknesses:**

1. Despite its apparent comprehensiveness, the empirical analysis lacks depth, especially considering the benchmark's cross-platform nature. The paper merely reports metrics without providing meaningful interpretation. It fails to offer an in-depth analysis of the key challenges posed by the cross-platform environment and the capabilities that best correlate with addressing these challenges.
2. The paper inadequately explains the construction process for graph evaluators. It provides a vague description of human involvement in the annotation process and lacks clear criteria for decomposing tasks into sub-tasks. Furthermore, there's limited explanation of how cross-platform dependencies are handled within the graph structure.
3. The evaluation framework is unnecessarily complex with multiple metrics (Success Rate, Completion Ratio, Execution Efficiency, Cost Efficiency). A single, well-designed metric would be more practical and interpretable.
4. (minor) The paper has several writing issues like inconsistent descriptions (e.g., "6 popular MLMs" vs "four advanced MLMs") and redundant entries in tables (e.g., repeated OSWorld entry in Table 1).

**Questions:**

1. What efforts were made to ensure CRAB's tasks reflect real-world scenarios? Can you provide a breakdown of task distributions by category?
2. How long does a single evaluation take?

---

> ### Author Response · Authors · 2024-11-24
> **Rebuttal by Authors**
>
> We sincerely thank the reviewer for the thorough and constructive comments. Please find the response to your questions below.
>
> > Empirical analysis lacks depth, especially considering the benchmark's cross-platform nature.
>
> In our study, the pipeline is quite complex, making it challenging to identify common issues that apply universally across all tasks and models. Based on our results, we observe that many of the challenges encountered in cross-platform tasks are not so different from those in single-platform settings. However, we did identify some key issues, particularly when analyzing the case study in Appendix C.3 and the termination reasons in Table 1. These include:
>
> **Action Space Discrepancies**: Cross-platform environments feature diverse action spaces across different platforms. This can confuse single-agent architectures, leading them to perform actions intended for one platform in another. In contrast, a multi-agent by environment setting avoids this issue, where the agent action spaces are divided by corresponding platforms.
>
> **Limited Context Length**: The models have limited context lengths, making it difficult to capture the entire trajectory of actions. This challenge is more severe in cross-platform settings, where the number of screenshots increases in each step, further limiting the agents' ability to process and adapt to unseen environments. WebRL [1] is exploring solutions to this issue by using reinforcement learning to tune model parameters.
>
> **Icon Recognition Failures**: Sometimes, the agents successfully identify the next action in text description but fail to recognize the correct icon to interact with. This issue may arise from a lack of icon-specific data in the training set, which is a good area for future research.
> Coordinate Grounding Issues in Complex Environments: While we used state-of-the-art technologies such as GroundingDINO and OCR, there are still occasional failures in detecting all screen elements. This problem is well-documented in the literature (e.g., Ferret-UI [3, 5], OmniParser [6], SeeClick [2], OS-ATLAS [4]).
>
> Additionally, we have provided a more detailed analysis of model-specific issues, including platform preferences, multi-agent versus single-agent performance, and termination reasons, in Appendix C.
>
> References:
>
> [1] Qi et al. “WebRL: Training LLM Web Agents via Self-Evolving Online Curriculum Reinforcement Learning.” arXiv, November 4, 2024.
>
> [2] Cheng et al. “SeeClick: Harnessing GUI Grounding for Advanced Visual GUI Agents.” arXiv, January 17, 2024.
>
> [3] Li et al. “Ferret-UI 2: Mastering Universal User Interface Understanding Across Platforms.” arXiv, October 24, 2024.
>
> [4] Wu et al. “OS-ATLAS: A Foundation Action Model for Generalist GUI Agents.” arXiv, October 30, 2024.
>
> [5] You et al. “Ferret-UI: Grounded Mobile UI Understanding with Multimodal LLMs.” arXiv, April 8, 2024.
>
> [6] Lu et al. “OmniParser for Pure Vision Based GUI Agent.” arXiv, July 31, 2024.

---

> > ### Author Response · Authors · 2024-11-24
> >
> > >  Clarification on graph evaluators.
> >
> > GDT is a flexible method, and the decomposition of tasks is designed to rely on the expertise and creativity of human annotators. However, we do provide foundational guidelines for our two-level task decomposition to ensure clarity and consistency. At the first level, where subtasks are decomposed into evaluators, our strategy is to ensure the decomposition is as detailed as possible while guaranteeing that all the resulting evaluators are **mandatory** for the task. For example, for a task requiring the agent to edit a file, the agent can use either a CLI editor such as Vim or a GUI text editor. The evaluator should not focus on the specific editor used but rather check whether the file has been edited.
> >
> > For the second-level decomposition, which involves breaking tasks into subtasks, we follow the principle that each subtask should **do one thing within a single environment**, with clearly defined inputs and outputs that enable seamless integration with other tasks. For example, downloading a file from a URL to a file path is a well-defined subtask: it accepts a URL as input and outputs the file’s contents. This subtask can connect to a task that outputs a URL or takes text contents as input as long as the types match, regardless of whether they originate from the same or a different platform. For these well-defined tasks, cross-platform dependency is similar to single-platform connections and does not require special consideration. For tasks with specific platform dependencies, we use LLMs to verify the connections and validate them further through human annotation.
> >
> > > Multiple metrics issue.
> >
> > While we understand the desire for simplicity, each metric in our framework serves a distinct purpose. Success Rate is a legacy metric used in most previous works, and we include it as a comparative method to highlight the importance of the Completion Ratio, which is our primary metric for evaluating agent performance. Our results demonstrate that the Completion Ratio provides a better assessment of agent performance, especially in cases where the Success Rate fails to reveal differences. Additionally, cost and efficiency are critical considerations for LLM agent users and researchers, making them important factors to address. To this end, we designed two secondary efficiency metrics for those interested in these aspects. Execution Efficiency evaluates the agent's ability to find optimal solutions with minimal errors, while Cost Efficiency addresses practical concerns about high inference costs, particularly in multi-agent setups.
> >
> > > Modifications regarding inconsistent descriptions.
> >
> > Thank you for pointing out. We will fix these issues in the revised version.

---

> > > ### Author Response · Authors · 2024-11-24
> > >
> > > > What efforts were made to ensure CRAB's tasks reflect real-world scenarios? Can you provide a breakdown of task distributions by category?
> > >
> > > In CRAB, all applications involved are widely used in real-world systems rather than artificially created applications specifically designed for the purpose, which could introduce biases. Our task set includes 25 real-world applications, which is very wide coverage in existing GUI agent benchmarks. This breadth presents significant challenges in designing effective evaluator functions across such a diverse range of applications. Therefore, our primary efforts focus on developing multiple evaluation methods to ensure that all tasks are evaluated accurately. These methods include parsing Android UI XML files, executing command-line commands, analyzing agent trajectories, and using advanced deep-learning models to assess image states. All evaluation methods are validated by human annotators to ensure correctness.
> > >
> > > Regarding task distribution, we provide a detailed list of the applications included in our dataset, along with the task count for each. Notice that a single application may be used for multiple tasks.
> > >
> > > | App Name        	| Description                                              	| # Tasks |
> > > | ------------------- | ------------------------------------------------------------ | ------- |
> > > | Terminal        	| GNOME terminal emulator, including many command line tools (cat, wget, mv, cp, etc.) | 40  	|
> > > | Firefox         	| Web browser, including various web applications (Google Docs, Google Search, etc.) | 35  	|
> > > | File Manager    	| GNOME official file manager                              	| 25  	|
> > > | GIMP            	| GNU Image Manipulation Program, free and open-source raster graphics editor | 13  	|
> > > | System Setting  	| GNOME system setting GUI application                     	| 11  	|
> > > | VSCode          	| Code editor                                              	| 8   	|
> > > | LibreOffice Writer  | Word processor                                           	| 8   	|
> > > | LibreOffice Impress | Presentation software                                    	| 7   	|
> > > | LibreOffice Calc	| Spreadsheet program                                      	| 6   	|
> > > | Vim             	| CLI text editor                                          	| 6   	|
> > > | Slack           	| Team communication platform                              	| 1   	|
> > >
> > > | App Name    	| Description                              	| # Tasks |
> > > | :-------------- | -------------------------------------------- | ------- |
> > > | Google Map  	| Map application                          	| 13  	|
> > > | Google Calendar | Calendar application                     	| 9   	|
> > > | Gmail       	| Google mail service application          	| 7   	|
> > > | Keep Notes  	| Google note application                  	| 6   	|
> > > | Google Tasks	| Google TO-DO list                        	| 5   	|
> > > | Messages    	| Android built-in message sending application | 5   	|
> > > | Contacts    	| Android built-in contacts application    	| 5   	|
> > > | Google Drive	| Google Cloud Drive application           	| 4   	|
> > > | Clock       	| Android built-in clock application       	| 2   	|
> > > | Files       	| Android built-in file manager            	| 1   	|
> > > | Settings    	| Android system setting                   	| 1   	|
> > > | Camera      	| Android built-in camera                  	| 1   	|
> > > | Google Docs 	| Google online word processor             	| 1   	|
> > > | Phone       	| Android built-in phone calling application   | 1   	|
> > >
> > > > How long does a single evaluation take?
> > >
> > > The duration of an evaluation varies and depends on the agent system structure, API response time, and the number of steps the agent takes to complete the task. On average, a single-agent system takes 10 to 20 seconds to complete one step, while a multi-agent system typically takes 20 to 40 seconds.

---

> > > > ### Comment · Reviewer_qvQK · 2024-11-25
> > > >
> > > > Thank you for the rebuttal. Having carefully reviewed the response, I would like to maintain my score.

---

> > > > > ### Author Response · Authors · 2024-11-25
> > > > >
> > > > > Thank you for your prompt response. Could you kindly let us know if there are any specific points raised in your review that we have not yet addressed? We would greatly appreciate the opportunity to provide further clarification to support your decision regarding the score.

---

### Official Review · Reviewer_9mhF · 2024-11-03

**Soundness:** 2
**Presentation:** 3
**Contribution:** 2
**Rating:** 3
**Confidence:** 4

**Summary:**

The paper introduces a benchmark framework  (CRAB) designed to evaluate the performance of MLM agents across different environments, such as desktop and mobile platforms. CRAB addresses the limitations of existing benchmarks by supporting cross-environment tasks, employing a graph-based fine-grained evaluation method, and offering an efficient mechanism for task and evaluator construction. The framework is extensible to any environment with a Python interface and includes a benchmark, Crab Benchmark-v0, consisting of 120 tasks across computer desktop and mobile phone environments. The study evaluates four advanced MLMs using various single and multi-agent system configurations, with GPT-4o achieving the highest completion ratio of 38.01%.

**Strengths:**

1. CRAB's ability to handle tasks across different environments is an advancement, reflecting the multi-platform nature of real-world applications.
2. The graph evaluator provides a nuanced and detailed assessment of agent performance, capturing intermediate progress and multiple valid pathways to task completion.
3.  The framework's design allows for easy adaptation to new platforms and devices, and its sub-task composition method streamlines task creation.
4. The inclusion of 120 real-world tasks in Crab Benchmark-v0 enhances the benchmark's applicability and usefulness for developing and testing MLM agents.
5. The paper presents a thorough evaluation of various MLMs and agent configurations, offering insights into their relative strengths and weaknesses.

**Weaknesses:**

This following comment reflects my personal bias, if the authors can convince me, I would be happy to change my score:

Although web agent recently is a hot topic, I feel that it is an engineering problem, not a research problem. In another words, as long we we gather enough web interaction data, all kinds of problems listed in those papers in the related works will not exist anymore. Therefore, just like 10 years ago, we spend quite a lot of effort in tweaking different models and benchmarks to solve vision and language problem and turn out of be useless, we should spend more time on studying how to scale up the training data if we want to solve the web agent problem. And even if it is solved, it won't teach us anything because the same lesson has learnt before. Therefore, it would be great for the author to elaborate on the broader lessons we can learn from reading this paper and how this problem can advance the AI field in general.

**Questions:**

Please see the comment in weakness section and please respond directly.

---

> ### Author Response · Authors · 2024-11-25
> **Rebuttal by Authors**
>
> Thank you for your thoughtful feedback and for highlighting your perspective on the nature of web agent research. While we acknowledge your concerns, we respectfully disagree with the notion that the development of web agents (or, in our case, cross-environment autonomous agents) is primarily an engineering challenge and not a research problem. Below, we address your points and elaborate on the broader contributions and lessons offered by this work:
>
> **Deep Learning Is Not Only About Data Scaling**
>
> While data scaling has been instrumental in advancing AI, it is not sufficient to address the complexities of all AI research. Methodology development is just as crucial as data scaling. For example, transformers [1] have enabled deep learning models to scale to unprecedented levels, forming the backbone of most large language models (LLMs). Similarly, diffusion models [2] have surpassed GANs [3] in a variety of vision tasks, even when operating on comparable data scales. These advancements highlight the importance of innovative methodologies in achieving stable and convincing results.
>
> **LLM Agents Are Not Just About Next-Token Prediction**
>
> LLMs excel in understanding the world and performing common reasoning tasks. The challenge LLM agents aim to address is how to better integrate LLMs with humans and the world without relying on collecting massive amounts of additional data. Should we manually collect large datasets every time a new task arises? Generalizing to unseen tasks and scenarios is a limitation that cannot be resolved solely by improving performance on existing data. This requires advancements in agent system designs or methodologies like reinforcement learning (RL). By enabling agents to interact with diverse environments and facilitating evaluation and data collection, CRAB addresses these inherently research-driven challenges. It aligns with the goals of frameworks like OpenAI Gym while significantly extending their capabilities, enabling the training of RL algorithms through exploration and interaction data collection.
>
> **CRAB as a Long-Term Benchmark**
>
> The core purpose of a benchmark is to evaluate the quality of a method—a role that remains unchanged despite technological advancements. For example, benchmarks like GLUE [4] have been instrumental in assessing natural language models over the years, while MMLU [5], introduced in 2020, continues to serve as one of the primary benchmarks in the LLM evaluation field. These examples highlight the enduring value of well-designed benchmarks. Similarly, CRAB is the first benchmark designed with a long-term vision to assess cross-environment autonomous agents, even though current LLM models are not yet capable of handling such complex real-world tasks well. Graph-based evaluation method used in CRAB offers detailed performance analysis, which provides nuanced metrics as valuable feedback to guide the bottleneck of LLM agent research. It also provides a scalable task creation method that can serve as an important mechanism for future benchmark development. Besides, by collaborating with a broad open-source community, we are committed to enhancing our benchmark in future iterations by incorporating a wider variety of task instances, supporting diverse platforms such as MacOS and Windows, and adopting more efficient evaluation methodologies.
>
> References:
>
> [1] Vaswani et al. "Attention is all you need." Advances in Neural Information Processing Systems 2017
>
> [2] Ho et al. "Denoising diffusion probabilistic models." Advances in Neural Information Processing Systems 2020
>
> [3] Goodfellow et al. "Generative adversarial networks." Communications of the ACM, 2020
>
> [4] Wang et al. "Glue: A multi-task benchmark and analysis platform for natural language understanding." International Conference on Learning Representations 2019
>
> [5] Hendrycks et al. "Measuring massive multitask language understanding." International Conference on Learning Representations 2021

---

> > ### Comment · Reviewer_9mhF · 2024-11-28
> >
> > Sorry for not clearly expressing my concerns. Just give some background, I have two students working on web agents and published multiple papers on top conferences. I also have a friend who has a company on RPA (Robotic Process Automation). After talking to my friend, I realized they their systems using relatively small models have very high accuracy or completion rate on some seemly difficult web agent problems. So I ask myself, why SOTA LLM like GPT-4o perform poorly on web agent tasks, is it because these tasks are difficult or because they have never seen such kind of data?
> >
> > So here are my suggestions: 1) can you check why GPT-4o perform poorly on your benchmark? 2) can you fine-tune LLLava to see how well they scale with training data? If a small amount of data can significantly increase the performance, then maybe it is just an out-of-domain problem.

---

> > > ### Author Response · Authors · 2024-11-29
> > >
> > > Thank you for your response and suggestions. Before addressing your questions individually, we would like to clarify the agent definition you referred to. As we noticed frequent mentions of **Web Agents** in your comments, we assume you are referring to **HTML-based websites running inside a browser** in the papers and products you cited. We would like to explain why our scenarios differ from "Web Agents" and highlight the distinctions. Please let us know if we have misunderstood your statement.
> > >
> > > To the best of our knowledge, most current Web Agents rely heavily on access to HTML source code, where all elements are clearly marked, and information is represented in a structured, text-based format. Additionally, this text-based structure is likely well-represented in LLM training corpora due to its ease of collection, allowing modern LLMs to process and understand such data effectively.
> > >
> > > In contrast, the environments we are working with—such as full desktop computers and mobile
> > > operating systems—pose significantly more complex challenges. These platforms lack a unified, text-based language to describe applications. Instead, text and interactive elements must be interpreted from **screenshots**, mimicking the perceptual process of a human observer. This reliance on visual understanding introduces a critical challenge for multimodal language models, as the agent lacks privileged access to system-level information like HTML source code. This remains a difficult task for both commercial and open-source models, even with fine-tuning.
> > >
> > > Furthermore, while we include tasks involving web apps in our dataset, the agent in our setup does not have access to the HTML source code and operates solely on screenshots as observations, making the setting more challenging than those typically addressed by Web Agents. Additionally, our dataset includes cross-platform tasks, further increasing the complexity of the required operations.
> > >
> > > With this clarification, we can address your questions:
> > >
> > > > Why GPT-4o perform poorly on your benchmark?
> > >
> > > GPT-4o already exhibits the best performance, showing a significant gap compared to other models. Additionally, its performance on our benchmark is not unusually poor when compared to other recent desktop and mobile benchmarks (e.g., GPT-4 achieves a 12.24% Success Rate (SR) in OS-World [1], GPT-4V-1106 achieves 19.5% SR in Windows Agent Arena [2], and GPT-4o achieves 31.16% in AndroidLab [3]). Upon analyzing case studies (Appendix C.3) and termination reasons (Table 1), we identified several key challenges within our benchmark (also common to other benchmarks):
> > >
> > > - Action Space Discrepancies: Cross-platform environments feature diverse action spaces across different platforms. This can confuse single-agent architectures, leading them to perform actions intended for one platform in another. In contrast, a multi-agent by environment setting avoids this issue, where the agent action spaces are divided by corresponding platforms.
> > >
> > > - Limited Context Length: The models have limited context lengths, making it difficult to capture the entire trajectory of actions. This challenge is more severe in cross-platform settings, where the number of screenshots increases in each step, further limiting the agents' ability to process and adapt to unseen environments. WebRL [1] is exploring solutions to this issue by using reinforcement learning to tune model parameters.
> > >
> > > - Icon Recognition Failures: Sometimes, the agents successfully identify the next action in text description but fail to recognize the correct icon to interact with. This issue may arise from a lack of icon-specific data in the training set, which is a good area for future research.
> > >
> > > - Coordinate Grounding Issues in Complex Environments: While we used state-of-the-art technologies such as GroundingDINO and OCR, there are still occasional failures in detecting all screen elements.
> > >
> > > > Can you fine-tune LLava to see how well they scale with training data?
> > >
> > > Existing research suggests that improving GUI agent performance is not straightforward. Unlike Web Agents, which can leverage system internals such as HTML source code, GUI agents often see minimal performance gains from small increases in training data. For instance:A small amount of additional training data does not necessarily result in significant performance gains. For example:
> > >
> > > * UGround [4] required 1.3M images to improve grounding accuracy from 25.6% to 46.8%, with the backend planner still relying on GPT-4o.
> > > * CogAgent [5], an 18B VLM specializing in GUI understanding and navigation, achieves only a 1.11% SR in OS-World, compared to GPT-4o's 12.24%.
> > >
> > > On the other hand, our paper is in **"dataset and benchmark"** area and primarily focuses on introducing an improved benchmark for evaluating multimodal language agents. While better model designs may emerge in the future, the exploration of specific design choices and experimental approaches lies beyond the scope of our work.

---

> > > > ### Author Response · Authors · 2024-11-29
> > > >
> > > > References:
> > > >
> > > > [1] Xie, Tianbao, et al. "Osworld: Benchmarking multimodal agents for open-ended tasks in real computer environments." arXiv preprint arXiv:2404.07972 (2024).
> > > >
> > > > [2] Bonatti, Rogerio, et al. "Windows agent arena: Evaluating multi-modal os agents at scale." arXiv preprint arXiv:2409.08264 (2024).
> > > >
> > > > [3] Xu, Yifan, et al. "AndroidLab: Training and Systematic Benchmarking of Android Autonomous Agents." arXiv preprint arXiv:2410.24024 (2024).
> > > >
> > > > [4] Gou, Boyu, et al. "Navigating the digital world as humans do: Universal visual grounding for gui agents." arXiv preprint arXiv:2410.05243 (2024).
> > > >
> > > > [5] Hong, Wenyi, et al. "Cogagent: A visual language model for gui agents." Proceedings of the IEEE/CVF Conference on Computer Vision and Pattern Recognition. 2024.

---

> ### Author Response · Authors · 2024-11-26
> **Kind reminder for review response**
>
> Dear Reviewer,
>
> We hope this message finds you well. We kindly request your feedback on our rebuttal comments. We would greatly appreciate it if you could review our response and engage in a constructive discussion with us.
>
> Best regards,
>
> The Authors

---

### Official Review · Reviewer_VWc7 · 2024-11-03

**Soundness:** 3
**Presentation:** 2
**Contribution:** 2
**Rating:** 3
**Confidence:** 3

**Summary:**

The paper introduces the CRAB framework, designed to evaluate multi-modal language models MLMs in a cross-platform setting. CRAB supports multi-agent configurations and uses a graph-based task decomposition approach to allow flexible, multi-path task completion across environments like desktop and mobile platforms. A unique feature is the graph-based evaluator that provides a detailed, granular assessment of task progress by tracking completion across multiple paths, which extends beyond traditional single-path evaluations. Experiments with commercial and open-source MLM models on Ubuntu and Android environments demonstrate CRAB’s potential for flexible multi-agent, cross-platform assessments.

**Strengths:**

A key contribution of the paper is the cross-platform evaluation capability, allowing MLMs to be assessed across desktop and mobile platforms. The graph-based task decomposition enables flexible, multi-path task completion, and the detailed evaluation metrics offer a comprehensive assessment of agent performance. The diverse experimental setup with various MLM models shows CRAB’s broad applicability.

**Weaknesses:**

1.	The paper lacks sufficient review of existing work, missing comparisons with related frameworks such as Mobile-ENV [1] and GUI-World [2].
2.	Table 1 only presents the number of apps, whereas similar papers like AndroidWorld [3] provide detailed counts of tasks and templates per app. Additionally, "osworld" appears twice in the table, which could cause confusion and detracts from a thorough task-level comparison.
3.	The explanation of GDT (Graph of Decomposed Tasks) is unclear; it appears more like a detailed "tree of thought" rather than a true graph structure, which may confuse readers regarding the intended data structure.
4.	The necessity of a multi-agent system is not well justified, as no significant performance improvements are shown over a single-agent configuration.
5.	The absence of a comparison with traditional trajectory-based evaluation methods limits the paper’s ability to showcase the distinct advantages of CRAB. Additionally, each task is based on a simple 4-node graph, raising questions about how significantly the results could differ from those of trajectory-based evaluation.
Reference:
[1] Zhang, Danyang, Lu Chen, and Kai Yu. 2023. "Mobile-ENV: A Universal Platform for Training and Evaluation of Mobile Interaction." arXiv preprint arXiv:2305.08144.
[2] Chen, Dongping, Yue Huang, Siyuan Wu, Jingyu Tang, Liuyi Chen, Yilin Bai, Zhigang He, Chenlong Wang, Huichi Zhou, Yiqiang Li, et al. 2024. "GUI-World: A Dataset for GUI-Oriented Multimodal LLM-Based Agents." arXiv preprint arXiv:2406.10819.
[3] Rawles, Christopher, Sarah Clinckemaillie, Yifan Chang, Jonathan Waltz, Gabrielle Lau, Marybeth Fair, Alice Li, William Bishop, Wei Li, Folawiyo Campbell-Ajala, et al. 2024. "AndroidWorld: A Dynamic Benchmarking Environment for Autonomous Agents." arXiv preprint arXiv:2405.14573.

**Questions:**

N/A

---

> ### Author Response · Authors · 2024-11-24
> **Rebuttal by Authors**
>
> We sincerely thank the reviewer for the thorough and constructive comments. Please find the response to your questions below.
>
> > Missing comparisons with related frameworks.
>
> We appreciate the reviewer’s suggestion to include comparisons with Mobile-ENV and GUI-World. In response, we will revise the paper to address this feedback by adding these comparisons in the related work section and updating Table 1 accordingly. Regarding Mobile-ENV, we reference our comparison on its most recent version (v4). Mobile-ENV introduces intermediate evaluation and instruction and proposes two datasets: the open-world set and wikiHow, specifically designed for mobile phone environments. While its combination of signals aligns with our graph evaluation approach, its reliance on a tree structure with multiple relationships can introduce redundancy and increase complexity for annotators, potentially limiting dataset scalability. As for GUI-World, it contributes a substantial video dataset for GUI automation and trains a new VideoLLM for UI tasks. However, its evaluation approach, LLM-as-a-Judger, may lack the precision and consistency offered by rule-based evaluation systems. We believe CRAB offers a more concise and scalable evaluation framework that could serve as a valuable tool for advancing UI automation in multimodal models.
>
> > Suggestions regarding Table 1.
>
> As suggested by the reviewer, we present the applications in our task dataset along with the counts of tasks that utilize them below. It is also worth noting that in our task settings, a single task often involves two or more applications. On average, each task contains 1.84 applications, according to our statistics. Apologies for the duplicated “OSWorld” in Table 1. We will change it in the revised version.
>
> | App Name        	| Description                                              	| # Tasks |
> | ------------------- | ------------------------------------------------------------ | ------- |
> | Terminal        	| GNOME terminal emulator, including many command line tools (cat, wget, mv, cp, etc.) | 40  	|
> | Firefox         	| Web browser, including various web applications (Google Docs, Google Search, etc.) | 35  	|
> | File Manager    	| GNOME official file manager                              	| 25  	|
> | GIMP            	| GNU Image Manipulation Program, free and open-source raster graphics editor | 13  	|
> | System Setting  	| GNOME system setting GUI application                     	| 11  	|
> | VSCode          	| Code editor                                              	| 8   	|
> | LibreOffice Writer  | Word processor                                           	| 8   	|
> | LibreOffice Impress | Presentation software                                    	| 7   	|
> | LibreOffice Calc	| Spreadsheet program                                      	| 6   	|
> | Vim             	| CLI text editor                                          	| 6   	|
> | Slack           	| Team communication platform                              	| 1   	|
>
> | App Name    	| Description                              	| # Tasks |
> | :-------------- | -------------------------------------------- | ------- |
> | Google Map  	| Map application                          	| 13  	|
> | Google Calendar | Calendar application                     	| 9   	|
> | Gmail       	| Google mail service application          	| 7   	|
> | Keep Notes  	| Google note application                  	| 6   	|
> | Google Tasks	| Google TO-DO list                        	| 5   	|
> | Messages    	| Android built-in message sending application | 5   	|
> | Contacts    	| Android built-in contacts application    	| 5   	|
> | Google Drive	| Google Cloud Drive application           	| 4   	|
> | Clock       	| Android built-in clock application       	| 2   	|
> | Files       	| Android built-in file manager            	| 1   	|
> | Settings    	| Android system setting                   	| 1   	|
> | Camera      	| Android built-in camera                  	| 1   	|
> | Google Docs 	| Google online word processor             	| 1   	|
> | Phone       	| Android built-in phone calling application   | 1   	|

---

> > ### Author Response · Authors · 2024-11-24
> >
> > > Clarification on GDT (Graph of Decomposed Tasks).
> >
> > The GDT is indeed a fundamental idea in our approach, and it is implemented as a real graph structure rather than resembling a "Tree of Thoughts." The graph structure in GDT serves a dual purpose: **evaluation and task generation**.
> >
> > 1. **First-Level Graph Structure for Subtasks: Evaluation**
> >
> >     Each subtask in GDT is represented as a real graph structure, where every node functions as a binary evaluator. This enables a detailed and highly adaptive evaluation process for individual subtasks, ensuring comprehensive while fine-grained assessment of intermediate outcomes. Each node in the graph is a function that outputs a boolean value, determining whether the current system state fulfills the target described by the node. The edge between two nodes indicates a dependency, where the node being pointed to depends on the completion of the node from which the edge originates. Thus, a node will not be evaluated until all the nodes it depends on have been completed. For example, as illustrated in Figure 2 of our paper, the evaluator function for “put all files in the same directory” will not execute until all HTML files are downloaded and the Python script has been completed. We will explain the advantages of our evaluator in more detail in answer to weakness 5: “Comparison between trajectory-based evaluation.”
> >
> > 2. **Second-Level Structure for Composite Tasks: Task Generation**
> >
> >     Composite tasks in GDT are constructed by connecting multiple subtasks into a higher-level graph structure. This second-level structure plays a critical role in generating new tasks dynamically, leveraging the relationships and dependencies among subtasks.
> >
> > 3. **Key Difference from "Tree of Thoughts"**
> >
> >     Unlike "Tree of Thoughts," which is a prompting method designed to guide agents in problem-solving, the GDT is not visible to the agent and is exclusively used for evaluation and task generation within the system. These two methods focus on totally different aspects, "Tree of Thoughts" focuses on improving the agent cognitive pipeline, while GDT focuses on how we can better evaluate agent performance in real-world applications.
> >
> >
> > > Clarification on the multi-agent system.
> >
> > Our goal, as a benchmark, is to provide a clear and comprehensive evaluation across different settings. With the growing popularity of multi-agent systems in the LLM agent field, we believe that including these systems as baseline results will help future researchers in designing more effective agent system structures. From our experiments on both single- and multi-agent setups, we observed that hallucinations or biases in the upstream agents can lead to misunderstanding by downstream agents. For a more detailed analysis of the multi-agent system and a comparison with single-agent systems, please refer to Appendix C.2. However, our primary focus is to offer an improved benchmark for evaluating multimodal language agents. While better design choices for our baseline multi-agent system may emerge in the future, detailed design choices and experiments fall outside the scope of our paper.

---

> > > ### Author Response · Authors · 2024-11-24
> > >
> > > > Clarification on traditional trajectory-based evaluation methods.
> > >
> > > We did not include quantitative results comparing the two approaches because the graph evaluator differs fundamentally from trajectory-based evaluation in concept and is suited to different scenarios, as discussed in Section 4.2.
> > >
> > > A key distinction between our evaluation approach and traditional trajectory-based evaluation lies in our ability to handle more flexible scenarios. For instance, in a downstream task like creating a poster, which requires two unrelated inputs (finding an image and writing the text). The order in which these inputs are completed does not impact the assessment. Similarly, in tasks with multiple valid approaches, such as copying files from one folder to another, an agent might use the `cp` command in a terminal or perform the task through several clicks in a file manager with UI. Additionally, the agent might require multiple attempts before finding the correct solution. Our DAG-based graph evaluation method accommodates such variations naturally.
> > >
> > > In contrast, traditional trajectory-based evaluation methods often consider operations performed in an order different from the gold sequences as incorrect. While we acknowledge that trajectory-based evaluation can be simple and effective in very straightforward scenarios, such as tasks requiring just one or two clicks, it becomes less suitable in more complex, cross-environment settings like ours.
> > >
> > > Besides, We would also like to clarify the statement: “each task is based on a simple 4-node graph.” In fact, our task dataset includes graphs with up to 11 nodes. The distribution of node counts is provided below. It is important to note that the number of nodes depends on the complexity of the task, with more complex tasks involving larger graphs.
> > >
> > > |# nodes| 1	| 2	| 3	| 4	| 5	| 6	| 7	| 8	| 9	| 10   | 11   |
> > > |----| ---- | ---- | ---- | ---- | ---- | ---- | ---- | ---- | ---- | ---- | ---- |
> > > |# tasks |5| 16 |29| 26| 14 |18 | 7|  1|  0|  3| 1 |

---

> ### Author Response · Authors · 2024-11-26
> **Kind reminder for review response**
>
> Dear Reviewer,
>
> We hope this message finds you well. We kindly request your feedback on our rebuttal comments. We would greatly appreciate it if you could review our response and engage in a constructive discussion with us.
>
> Best regards,
>
> The Authors

---

> ### Author Response · Authors · 2024-12-02
> **Follow-Up on Rebuttal Discussion**
>
> Thank you once again for your valuable feedback. With less than two days remaining in the discussion period, we wanted to kindly follow up to check if you have any additional questions or concerns regarding our rebuttal.
>
> We have worked to address all your points thoroughly and are happy to provide further clarification if needed. We also hope you might consider revisiting your score in light of the updates and responses we’ve provided.
>
> We sincerely appreciate your time and thoughtful review.

---

### Official Review · Reviewer_3JLv · 2024-11-04

**Soundness:** 3
**Presentation:** 3
**Contribution:** 3
**Rating:** 6
**Confidence:** 4

**Summary:**

This paper introduces CRAB, a novel benchmark framework for evaluating multimodal language model (MLM) agents on cross-environment tasks. CRAB supports multiple devices, incorporates a graph-based fine-grained evaluation method, and provides an efficient mechanism for constructing tasks and evaluators. The authors also present Crab Benchmark-v0, comprising 120 tasks across computer desktop and mobile phone environments. They evaluate four advanced MLMs in single and multi-agent configurations on this benchmark. The results show that a single agent with GPT-4o achieves the best completion ratio of 38.01%. It highlights the need for more effective autonomous agents.

**Strengths:**

Originality & Quality: CRAB is the first benchmark to incorporate cross-environment tasks, reflecting real-world scenarios. Novel graph evaluator and sub-task composition methods address the limitations of existing evaluation methods and benchmarks.

Clarity: Contain detailed descriptions of framework design, task dataset, agent implementations. Visual case studies provide examples to illustrate agent performance on specific tasks. Good analysis of results by platform, model, agent structure, and termination reasons.

Significance: This work addresses key limitations in previous methods, including platform diversity, evaluation metrics.
It enables standardized evaluation of MLM agents on complex real-world cross-platform tasks. Modular and extensible framework design allows easy adaptation to new platforms and tasks.

**Weaknesses:**

- Limited coverage of applications. It focuses on original apps in Ubuntu & Android on Pixel devices. Expanding to more apps and devices would further improve real-world coverage. The number of data instances (120 tasks) is relatively small. While the tasks are designed to be more complex than those in other benchmarks, extending the dataset to around 500 instances would enable more comprehensive and statistically significant comparisons between different models and agent settings. A larger dataset would better test the generalization abilities of the agents and increase the overall significance of the benchmark. I am willing to increase the score if the authors can consider it.

- Visual prompting methods do not fully recognize all interactive elements, potentially hurting agent performance. More advanced visual understanding techniques could be explored.

- Lack of human performance baselines makes it difficult to assess how far MLM agents are from human-level task completion abilities.

**Questions:**

- Can the authors comment on the costs associated with running the benchmark?

---

> ### Author Response · Authors · 2024-11-24
> **Rebuttal by Authors**
>
> We sincerely thank the reviewer for the thorough and constructive comments. Please find the response to your questions below.
>
> > Limited coverage of applications.
>
> We agree with the reviewer that expanding the dataset to include a broader range of applications, devices, and task instances is essential. While the current dataset size is constrained by the time-intensive nature of commercial API calls, resource limitations, and the complexity of testing multiple backend models and agent structures, we are actively addressing these challenges as part of a long-term project. In collaboration with the broader open-source community, we are committed to enhancing the dataset in future iterations by incorporating a wider variety of task instances, supporting diverse platforms such as MacOS and Windows, and adopting more efficient evaluation methodologies. However, given the time constraints for the rebuttal, the suggestion to expand the tasks to 500 instances, unfortunately, cannot be easily achieved. This is because additional human effort is required to carefully verify the correctness of task semantics and the evaluation process, even though parts of the procedures used to create new tasks are already automated.
>
> > Visual prompting methods do not fully recognize all interactive elements, potentially hurting agent performance. More advanced visual understanding techniques could be explored.
>
> We acknowledge the limitations of the current visual prompting methods, despite the GroundingDINO + EasyOCR setting performance being good enough in our agent settings to detect all the required elements according to our observation. To address this, we are actively conducting experiments with more advanced visual understanding techniques such as OmniParser [1] and Ferret-UI [2, 3] and aim to report preliminary results by the end of the discussion period.
>
> > Lack of human performance baselines makes it difficult to assess how far MLM agents are from human-level task completion abilities.
>
> We recognize the importance of including human performance baselines. We are currently working on experiments and will provide results within the discussion period as well.
>
> > Costs associated with running the benchmark.
>
> Here is a table of estimated cost:
>
> | Model       	| Structure 	| Approx. USD per task |
> | --------------- | ------------- | -------------------- |
> | GPT-4o      	| Single    	| $0.25            	|
> | GPT-4o      	| Multi-by-func | $0.25            	|
> | GPT-4o      	| Multi-by-env  | $0.45            	|
> | GPT-4 Turbo 	| Single    	| $1.2             	|
> | GPT-4 Turbo 	| Multi-by-func | $1.2             	|
> | GPT-4o (w/o FC) | Single    	| $0.2             	|
> | Claude 3 Opus   | Single    	| $2               	|
> | Claude 3 Opus   | Multi-by-func | $2               	|
>
> References:
>
> [1] Lu, Yadong, Jianwei Yang, Yelong Shen, and Ahmed Awadallah. “OmniParser for Pure Vision Based GUI Agent.” arXiv, July 31, 2024.
>
> [2] Li, Zhangheng, Keen You, Haotian Zhang, Di Feng, Harsh Agrawal, Xiujun Li, Mohana Prasad Sathya Moorthy, Jeff Nichols, Yinfei Yang, and Zhe Gan. “Ferret-UI 2: Mastering Universal User Interface Understanding Across Platforms.” arXiv, October 24, 2024.
>
> [3] You, Keen, Haotian Zhang, Eldon Schoop, Floris Weers, Amanda Swearngin, Jeffrey Nichols, Yinfei Yang, and Zhe Gan. “Ferret-UI: Grounded Mobile UI Understanding with Multimodal LLMs.” arXiv, April 8, 2024.

---

> ### Author Response · Authors · 2024-11-26
> **Kind reminder for review response**
>
> Dear Reviewer,
>
> We hope this message finds you well. We kindly request your feedback on our rebuttal comments. We would greatly appreciate it if you could review our response and engage in a constructive discussion with us.
>
> Best regards,
>
> The Authors

---

> > ### Comment · Reviewer_3JLv · 2024-11-29
> > **Thanks for the response**
> >
> > Thanks for the responses! They addressed most of my questions and I would like to keep my current rating.

---

> ### Author Response · Authors · 2024-12-02
> **Additional experimental results and analysis**
>
> We sincerely thank the reviewer for their thorough and constructive comments. The reviewer raised some concerns regarding the visual prompting method and human performance. In response, we have provided additional experimental results and analysis, which we hope will address the reviewer's concerns.
>
> ## Human Performance
>
> Given that our task requires some level of Linux knowledge, we conducted human experiments with four undergraduate students: two majoring in computer science and two from other majors. This was done to ensure the results reflect an average performance level. The participants were divided into two groups based on their majors, and each participant completed 60 tasks randomly selected from the dataset.
>
> The results show that the overall success rate (SR) was 75.00%, and the completion ratio was 85.10%. These findings suggest that the difficulty of our tasks aligns with existing benchmarks (e.g., 78.24% SR for WebArena [1], 72.36% SR for OSWorld [2], and 80% SR for AndroidWorld [3]). Moreover, the results highlight a significant performance gap between MLMs and humans. The types of tasks at which humans excel differ from those where MLMs perform well. Specifically, MLMs are proficient at tasks requiring Linux-specific knowledge, such as command-line operations, while humans are better at tasks involving graphical operations, such as image editing and web-related activities.
>
> This highlights a limitation in the training data for MLMs, which appears to lack sufficient examples of graphical operations. Additionally, the reasoning ability of MLMs is still inadequate for inferring solutions to unseen graphical tasks, which humans find relatively easy.
>
> ## Advanced Visual Prompt Techinques
>
> We attempted to use OmniParser [4] and Ferret-UI 2 [5] as replacements for the Grounding Dino and EasyOCR visual prompt techniques in our pipeline. However, neither showed better performance within our framework.
>
> For OmniParser, we tested it on a subset of our dataset. The results indicate that the agent generates more invalid actions, many of which involve clicking on elements not provided on the screen. This issue may stem from differing filtering strategies. OmniParser retains most overlapping detections, which does not significantly affect scenarios where elements are provided in text form since the agent can identify them even with multiple overlaps. However, in our pipeline, elements are provided only through the Set-of-Marks, and overlapping elements tend to confuse the agent, leading to incorrect clicks. Additionally, OmniParser often detects more elements than our method due to its filtering strategy. While this might seem advantageous, the additional elements can further confuse the agent in our pipeline.
>
> Ferret-UI 2, on the other hand, exhibited severe issues, frequently generating a large amount of meaningless duplicated information, even when tested on its own example testcase (e.g., this [demo](https://huggingface.co/spaces/jadechoghari/ferret-demo)).
>
> For instance, one test case output was as follows:
>
> > The screen contains a variety of elements. At the top, there's a button labeled 'Apple' and a text that reads 'Reminders, Don't forget. Use reminders.' Below this, there are three buttons labeled 'OPEN', '4+, Categories, up', and '4+, Activity, up'. There's also a picture with a caption '200K Ratings, 4+, Activity, 4+, Activity, 4+, Activity, 4+, Activity, 4+, Activity, 4+, Activity, 4+, Activity, 4+, Activity, 4+, Activity, 4+, Activity, 4+, Activity, 4+, Activity, 4+, Activity, 4+, Activity, 4+, Activity, 4+, Activity, 4+, Activity, 4+, Activity, 4+, Activity, 4+, Activity, 4+, Activity, 4+, Activity, 4+, Activity, 4+, Activity, 4+, Activity, 4+, Activity, 4+, Activity, 4+, Activity, 4+, Activity, 4+, Activity, 4+, Activity, 4+, Activity, 4+, Activity, 4+, Activity, 4+, Activity, 4+, Activity, 4+, Activity, 4+, Activity, 4+, Activity, 4+, Activity, 4+, Activity, 4+, Activity, 4+, Activity, 4+, Activity, 4+, Activity, 4+, Activity, 4+, Activity, 4+, Activity, 4+, Activity, 4+, Activity, 4+, Activity, 4+, Activity, 4+, Activity, 4+, Activity, 4+, Activity, 4+, Activity, 4+, Activity, 4+, Activity, 4+, Activity, 4+, Activity, 4+, Activity, 4+, Activity, 4+, Activity, 4+, Activity, 4+, Activity, 4+, Activity, 4+, Activity, 4+, Activity, 4+, Activity, 4+, Activity, 4+, Activity, 4+, Activity, 4+, Activity, 4+, Activity, 4+, Activity, 4+, Activity, 4', and a text that reads 'Today'. There's also a picture with a caption '200K Ratings, 4+, Activity, 4+, Activity, 4+, Activity, 4+, Activity, 4+, Activity,
>
> Ferret-UI 2 also focuses exclusively on understanding mobile screenshots, which does not align with the cross-platform requirements of our pipeline. As such, we concluded that this model is unsuitable for integration into our system.

---

> ### Author Response · Authors · 2024-12-02
> **References**
>
> [1] Zhou, Shuyan, et al. "Webarena: A realistic web environment for building autonomous agents." arXiv preprint arXiv:2307.13854 (2023).
>
> [2] Xie, Tianbao, et al. "Osworld: Benchmarking multimodal agents for open-ended tasks in real computer environments." arXiv preprint arXiv:2404.07972 (2024).
>
> [3] Rawles, Christopher, et al. "AndroidWorld: A dynamic benchmarking environment for autonomous agents." arXiv preprint arXiv:2405.14573 (2024).
>
> [4] Lu, Yadong, et al. "Omniparser for pure vision based gui agent." arXiv preprint arXiv:2408.00203 (2024).
>
> [5] Li, Zhangheng, et al. "Ferret-ui 2: Mastering universal user interface understanding across platforms." arXiv preprint arXiv:2410.18967 (2024).

---

### Author Response · Authors · 2024-11-27

We greatly appreciate all the constructive comments from the reviewers. We have revised our paper according to the suggestions. The updates are as follows:

- We added all the mentioned related works by **Reviewer VWc7** to Table 1 and Section 2 "Related Work".
- We reorganized and provided more concrete explanations in Sections 4.2 "Graph Evaluator" and 4.4 "Task and Evaluator Construction" to clarify the construction process for graph evaluators, as requested by **Reviewer qvQK**.
- We added estimated single evaluation time according to the question raised by **Reviewer qvQK**.
- We added a paragraph "Key issues in solving cross-environment task" in Section 6.2 "Result Analysis" and revised other paragraphs in this section to remove redundant statements, as requested by **Reviewer qvQK**.
- We added Appendix A.1 "Dataset Statistics" to provide task distributions by application and evaluator nodes counts distribution, as suggested by  **Reviewer VWc7** and **Reviewer qvQK**.
- We addressed the minor text issues as noted by the reviewers.

We have highlighted all the revised/added parts with blue color for the reviewers' convenience. We will remove the colorization in the formal version. We are also looking forward to reviewers' feedback on this revision.

---

### Meta-Review · Area_Chair_oVQd · 2024-12-20

**Metareview:**

This paper introduced a new agent benchmark framework called CRAB for cross-environment tasks. The framework supports a graph-based evaluation method and supports multiple devices (desktops, phones). The authors used CRAB to evaluate advanced MLMs, including GPT4o. However there are some major concerns in the current work: (1) task coverage: the number of tasks is quite small and the tasks are limited to certain setups (2) lack of comparison to related work and analysis of the experimental results (3) due to the limited number of tasks, the current results might not be ideal to demonstrate the robustness of model agents (though I appreciate the authors provided the additional human evaluation results).

**Additional Comments On Reviewer Discussion:**

n/a

---

### Decision · Program_Chairs · 2025-01-22

Reject